



# 1  Quantifying methane emissions from natural gas production in
# 2  northeastern Pennsylvania

Zachary R. Barkley[1], Thomas Lauvaux[1], Kenneth J. Davis[1], Aijun Deng[1], Yanni Cao[1], Colm Sweeney[2],
Douglas Martins[7], Natasha L. Miles[1], Scott J. Richardson[1], Thomas Murphy[4], Guido Cervone[5], Anna
Karion[2], Stefan Schwietzke[8], MacKenzie Smith[3], Eric A. Kort[3], Joannes D. Maasakkers[6]
[1]Department of Meteorology, The Pennsylvania State University, University Park, PA 16802, United States
[2]NOAA/Earth Systems Research Laboratory, University of Colorado, Boulder, CO, 80305, United States
[3]Department of Climate and Space Sciences and Engineering, University of Michigan, Ann Arbor, MI, 48109, United States
[4]Marcellus Center for Outreach and Research, The Pennsylvania State University, University Park, PA 16802, United States
[5]Department of Geography, The Pennsylvania State University, University Park, PA 16802, United States
[6]School of Engineering and Applied Sciences, Harvard University, Pierce Hall, 29 Oxford Street, Cambridge, Massachusetts
02138, United States
[7]FLIR Systems, West Lafayette, IN 47906, United States
[8]Cooperative Institute for Research in Environmental Sciences, University of Colorado, Boulder, Colorado, USA.

*Correspondence to*: Zachary R. Barkley (zrb5027@psu.edu)
**Abstract.** Natural gas infrastructure releases methane ($CH_4$), a potent greenhouse gas, into the atmosphere. The estimated
emission rate associated with the production and transportation of natural gas is uncertain, hindering our understanding of its
greenhouse footprint. This study presents a new application of inverse methodology for estimating regional emission rates
from natural gas production and gathering facilities in northeastern Pennsylvania.  An inventory of $CH_4$ emissions was
compiled for major sources in Pennsylvania. This inventory served as input emission data for the Weather Research and
Forecasting model with chemistry enabled, and atmospheric $CH_4$ mole fraction fields were generated at 3km resolution.
Simulated atmospheric $CH_4$ enhancements from WRF-Chem were compared to observations obtained from a three-week flight
campaign in May 2015. Modelled enhancements from sources not associated with upstream natural gas processes were
assumed constant and known and therefore removed from the optimization procedure, creating a set of observed enhancements
from natural gas only. Simulated emission rates from unconventional production were then adjusted to minimize the mismatch
between aircraft observations and model-simulated mole fractions for ten flights. To evaluate the method, an aircraft mass
balance calculation was performed for four flights where conditions permitted its use. Using the model optimization approach,
the weighted mean emission rate from unconventional natural gas production and gathering facilities in northeastern
Pennsylvania approach is found to be 0.36% of total gas production, with a 2σ confidence interval between 0.27-0.45% of
production. Similarly, the mean emission estimates using the aircraft mass balance approach is calculated to be 0.34% of
regional natural gas production, with a 2σ confidence interval between 0.06-0.62% of production. These emission rates as a



percent of production are lower than rates found in any other basin using a top-down methodology, and may be indicative of
some characteristics of the basin that makes sources from the northeastern Marcellus region unique.

## 1 Introduction

The advent of hydraulic fracturing and horizontal drilling technology has opened up the potential to access vast reservoirs of
natural gas previously inaccessible, shifting energy trends in the United States away from coal and towards natural gas (EIA,
2016b). From a greenhouse gas (GHG) emissions perspective, natural gas has the potential to be a cleaner energy source than
coal. For every unit of energy produced, half as much carbon dioxide ($CO_2$) is emitted through the stationary combustion of
natural gas in comparison to coal (EPA, 2016). However, during the process of extracting and distributing natural gas a
percentage of the overall production escapes into the atmosphere through both planned releases and unintended leaks in
infrastructure. Though these emissions may be from an economic perspective, their climatological impacts are not negligible
(Alvarez et al., 2012; Schwietzke et al., 2014). Methane ($CH_4$), the main component of natural gas, is a potent greenhouse gas
with a global warming potential over a 20-year period ($GWP_{20}$) of 84 (Myhre et al., 2013). Over a 100-year period the GWP
is reduced to 28 due to interactions with the hydroxyl radical which transform the $CH_4$ molecule to $CO_2$. Depending on which
timespan is used, the relative climatological impacts of natural gas as an energy source compared to coal can vary. Using the
$GWP_{20}$ value, it is estimated that a natural gas emission rate of only 3% of total gas production would result in a natural gas
power plant having a more negative impact on the climate than a coal-powered plant. Using the $GWP_{100}$ value, this emission
rate threshold shifts to 10% of production (Schwietzke et al., 2014; Alvarez et al., 2012). Complicating matters further, the
future climate impacts associated with an increased availability of natural gas extends well beyond a simple greenhouse gas
footprint comparison against coal. Lower fuel prices linked to this new reservoir of energy can change the course of future
energy development globally. With many states and countries attempting to find a suitable balance between their energy
policies and greenhouse gas footprint, it is important for the scientific community to be able to quantify and monitor natural
gas emission rates.
The mining and transfer of natural gas can be broken down into five stages: production, processing, storage,
transmission, and distribution. The Environmental Protection Agency (EPA) uses a bottom-up approach to quantify these
emissions, estimating emission rates per facility or component (such as a compressor, unit length of pipeline, pneumatic device)
or an average emission per event (such as a well completion or liquids unloading). These "emission factors" are then multiplied
by nationwide activity data containing the number of components or events associated with each emission factor, and a total
emission rate is produced for the country (EPA, 2015b). This bottom-up approach is a practical methodology for estimating
emissions over a large scale but has limitations. A bottom-up inventory depends on the quality and quantity its emission factors
and activity data. Emissions from sources in the natural gas industry can be temporally variable and have a wide range of
values depending on a number of factors, such as the quality and age of the device and the gas pressure moving through the
component. Furthermore, recent studies have shown that a majority of emissions comes from a small percentage of devices,



often referred to as "super-emitters", creating a long-tail distribution of emission sources (Brandt et al., 2014, Omara et al.,
2016, Zavala-Araiza et al., 2015, 2017, Frankenberg et al., 2016). These two factors make it difficult to sample enough devices
and adequately describe the mean emission rate, thus allowing for significant representation errors in the emission factors.
Because emission factors are required for hundreds of different components, these errors can accumulate and lead to systematic
biases in the total emissions estimate.
One way to compare results based on inadequate sample sizes in the bottom-up approach is to measure the aggregated
enhancement in the atmospheric mole fraction at larger scales through a top-down approach. Instead of measuring emissions
from individual devices and scaling up, a top-down approach takes atmospheric greenhouse gas concentrations measured
downwind of a continent (e.g. Bousquet et al., 2006), a region (e.g. Lauvaux et al., 2008), a city (e.g. White et al., 1976, Mays
et al., 2009, Lamb et al., 2016) or a facility (e.g. Ryerson et al., 2001) and uses inverse methodologies to attribute the
enhancements to potential sources upwind. One of these methods, the aircraft mass balance technique, has been performed at
many different oil and gas fields to characterize natural gas emissions (Petron et al., 2012, Karion et al., 2013, 2015, Peischl
et al., 2015, Conley et al., 2016). While this methodology is able to capture surface fluxes over a large region, it remains
difficult to attribute the emissions to any individual source (Cambaliza et al., 2014). Any sources from within the flux region
that emit $CH_4$ will be measured in the downwind observations and be a part of the aggregated regional enhancement.
Atmospheric observations also include other sources unrelated to natural gas, such as anaerobic respiration from landfills and
wetlands, enteric fermentation from cattle, anaerobic decomposition of manure, $CH_4$ seepage from coal mining, and many
other smaller sources. If the purpose of the study is to solve for the emissions from the natural gas industry, emissions from all
sources unrelated to natural gas must be known and removed from the regional flux estimate. Thus, top-down experiments
require an accurate $CH_4$ inventory of the study area and any errors associated with the inventory will propagate into the final
emissions estimate. A more advanced technique to separate out non-natural gas sources has been developed using ethane as a
tracer for natural gas (Smith et al., 2015). However, such methods may struggle in dry gas basins where smaller ethane to
methane ratios within the gas can make the ethane signature more difficult to separate out. And similar to bottom-up methods,
top-down studies fail to address temporal variability, with observations from many of these studies having been collected
during a limited number of 2 to 4 hour aircraft flights performed over a period of weeks.
In recent years, both bottom-up and top-down studies have aimed at calculating natural gas emission rates, with
bottom-up studies generally finding smaller emission rates than their top-down counterparts (Brandt et al., 2014). The
discrepancy between the results from these two methodologies must be better understood if the true emission rate is to be
known. Both the bottom-up and top-down approaches have their own inherent sources of error. For the bottom-up approach,
a small sample size could result in the omission of any super-emitters, resulting in a low emissions bias. For the top-down
approach, difficulty in attributing the measured enhancements to their correct sources can lead to errors when solving for the
emissions of a particular sector.
Top-down emission estimates of individual basins have shown variation in the emission rate across the different
basins. An aircraft mass balance performed over the Barnett shale in Texas found an emission rate between 1.3-1.9% of



production (Karion et al., 2015), yet a similar mass balance study executed over unconventional wells in Uintah County, Utah,
calculated an emission rate between 6.2-11.7% of production (Karion et al., 2013). Differences in regional emission rates can
perhaps best be illustrated by recent studies in the Marcellus region. The Marcellus shale gas play is part of the Marcellus
geological formation running close to the Appalachian mountain chain from West Virginia to southern New York and contains
an estimated 140 billion cubic feet of technically recoverable natural gas (EIA, 2012). Reaching peak production by the end
of 2015, the Marcellus is the largest producing shale in the U.S., producing 17,000 million standard cubic feet (MMSCFD)
per day of natural gas (EIA, 2016a). A bottom-up study measuring emissions from 17 unconventional well-sites in the
Marcellus found a median emission rate from the wells of 0.13% of production, but estimated a mean emission rate of 0.53%
of production due to the potential presence of super-emitters which would skew the mean emission rate towards values higher
than the median (Omara et al., 2016). An aircraft mass balance study over northeastern Pennsylvania calculated an emission
rate of 0.30%, a number that accounted for emissions from the production, processing, and transmission of the gas (Peischl et
al., 2015). Both of these derived estimates fall below emission rates calculated throughout other basins and are below the 3%
threshold required for natural gas to be a smaller climate pollutant in comparison to coal over a 20-year timescale. The low
rates in the Marcellus compared to other regions could be the result of systematic difference within the Marcellus that leads to
a more efficient extraction of natural gas. However, while useful as a first-guess estimation, current studies performed in the
region are based on relatively small sample sizes (1 aircraft mass balance and 88 individual well measurements). A more
thorough analysis of the emission rate in the Marcellus would provide insight into regional differences in $CH_4$ emissions from
different shale basins and help improve national estimates of emissions from natural gas.

This study seeks to provide confidence in the emission rate for the northeastern Marcellus by performing the most

thorough top-down analysis of the northeastern Marcellus region to date. $CH_4$ measurements were taken from aircraft
observations across 10 flights in northeastern Pennsylvania. A new implementation of modelling $CH_4$ mole fractions is
developed to track complex plume structures associated with different emitters, and an optimal natural gas emission rate is
solved for each of the 10 flights. An aircraft mass balance technique is also conducted for 4 of the flights and natural gas
emission estimates from this method are compared to those calculated using the modelling technique. Using information on
the uncertainty with both methods, a regional emission rate is calculated for the natural gas industry in the northeastern
Marcellus region.

## 2 Methods

The objective of this study is to quantify $CH_4$ emissions coming from unconventional wells and compressor stations, henceforth
referred to as upstream natural gas emissions, in the northeastern Marcellus region (defined as the area contain within 41.1-
42.2°N 75.2-77.6°W, Figure 1) through two different top-down methodologies. $CH_4$ observations from aircraft data are
collected for ten (10) individual flights over a three-week period in May 2015. These data are used to solve for the upstream
natural gas emission rate using an aircraft mass balance approach. Additionally, a $CH_4$ emissions inventory for the region is



compiled and input into an atmospheric transport model described below. $CH_4$ concentrations are modelled for each flight,
and the upstream natural gas emission rate within the model is optimized to create the best match between aircraft observations
and model projected enhancement, providing another estimate for the upstream natural gas emission rate. The sections below
detail the regional $CH_4$ inventory, the aircraft campaign, the transport model, the model optimization technique, and the mass
balance approach used in this study.

**2.1 Regional Methane Emission Inventory**

In this study we characterize emissions from the natural gas industry into five different sectors: emissions from wells, emissions
from compressor facilities, emissions from storage facilities, emissions from pipelines, and emissions in the distribution sector.
To estimate $CH_4$ emissions from the production sector of the natural gas industry, data were first obtained on the
location and production rate of each unconventional well from the Pennsylvania Department of Environmental Protection Oil
and Gas Reporting website (PADEP, 2016) and the West Virginia Department of Environmental Protection (WVDEP, 2016).
To convert the production rate into an emission rate, we need to assume a first-guess as to the expected leakage from wells in
the area. A first-guess natural gas emission rate of 0.13% was applied to the production value of each of the 7000+ producing
unconventional wells based on the median rate from Omara et al., (2016). The natural gas emission rate was then converted to
a $CH_4$ emission rate by assuming a $CH_4$ composition in the natural gas of 95% (Peischl et al., 2015).
In addition to unconventional wells, the domain also contains more than 100,000 shallow conventional wells. Annual
conventional production rates for the year 2014 were obtained through the PADEP Oil and Gas Reporting website, the
WVDEP, and the New York Department of Environmental Conservation (NYDEC, 2016). Despite the large number of wells,
the average conventional well in PA produces 1% of the natural gas compared to its unconventional counterpart. However, it
is speculated that the older age of these wells and a lack of maintenance and care for them results in a higher emission rate for
these wells as a function of their production (Omara et al., 2016). A first-guess natural gas emission rate of 11% was applied
to the production values of the conventional wells based on the median emission rate from the wells sampled in Omara et al.,
(2016). Similar to the unconventional wells, the natural gas emission rate was then converted to a $CH_4$ emission rate by
assuming a $CH_4$ composition in the natural gas of 95%.
Compressor stations located within the basin are responsible for collecting natural gas from multiple well locations,
removing non-$CH_4$ hydrocarbons and other liquids from the flow, and regulating pressure to keep gas flowing along gathering
and transmission pipelines, and can be a potential source for methane emissions. Data for compressor station locations and
emissions comes from a dataset used in Marchese et al., (2015). A total of 489 compressor facilities are listed for Pennsylvania,
with 87% of the listed facilities also containing location data. Emissions for each compressor station are calculated through
two different methodologies. In the simplest case, a flat emission rate of 32.35 kg hr$^{-1}$ is applied for each station, the mean
emission rate of a gathering facility in PA found in Marchese et al., (2015). In the more complex scenario, the same emissions
total is used as in the flat rate case, but is distributed among the compressor stations linearly as a function of their energy usage.



Wattage between compressors in our dataset can vary greatly, from 10 kW for small compressors to 7000 kW or more at large
gathering facilities. Using the wattage as a proxy for emissions allows us to account for the size and throughput of natural gas
at each station and assumes larger stations will emit more natural gas compared to smaller stations (Marchese et al., 2015).
Data on locations of underground storage facilities were obtained from the United States Energy Information
Administration (EIA, 2015). For each of these locations, a base emission rate of 96.7 kg hr$^{-1}$ was applied according to the
average value emitted by a compressor station associated with an underground storage facility (Zimmerle et al., 2016).
To calculate pipeline emissions, data on pipeline locations needed to be collected. Information on transmission
pipelines, which connect gathering compressors to distribution networks, is provided by the Natural Gas Pipelines GIS product
purchased from Platts, a private organization which collects and creates various infrastructural layers for the natural gas and
oil industry (Platts, 2016). Gathering pipeline data, corresponding to the transfer of gas from wellheads to gathering
compressors, is nearly non-existent for PA with the exception of Bradford County, which maps out all gathering pipeline
infrastructure within the county border. In PA, information on the location of a gathering pipeline elsewhere is only available
where a gathering line crosses a stream or river. To account for gathering pipelines in the remainder of the state, a GIS model
was created using Bradford County as a typical pattern to simulate connecting pipelines between unconventional wells
throughout the state (Figure 2). The resulting pattern follows the valley of the Appalachian Mountains, with larger pipelines
crossing through the state to connect the different branches of the network. These pipelines were then multiplied by an emission
factor of 0.043 kg per mile of pipe, the factor used for gathering pipeline leaks in the Inventory of U.S. Greenhouse Gas
Emissions and Sinks: 1990-2013 (EPA, 2015b).
CH$_4$ emissions from natural gas distribution sources, coal mines, and animal/animal waste were provided from
Maasakkers et al., (2016), which takes national scale emissions from the EPA's greenhouse gas inventory for the year 2012
and transforms it into a $0.1° \times 0.1°$ emissions map for the continental U.S. For natural gas distribution emissions, various
pipeline data was collected at a state-level and emission factors were accounted for to calculate a total distribution emission
for the state. This emissions total was then distributed within the state proportional to the population density. Emission
estimates for coal are calculated using information from the Greenhouse Gas Reporting Program (GHGRP) for active mines
and the Abandoned Coal Mine Methane Opportunities Database for abandoned mines (EPA, 2008). State-level emissions
missions from enteric fermentation and manure management are provided in the EPA's inventory. These emissions were
segregated into higher resolutions using county-level data from the 2012 U.S. Census of Agriculture (USDA, 2012) and land-
type mapping.
Finally, the EPA's Greenhouse Gas Reporting Program dataset for the year 2014 was used to capture all other major
sources of CH$_4$ in the region otherwise unaccounted for, the majority of which are emissions from landfills and some industrial
sources (EPA, 2015a). Sources within the GHGRP that overlap with natural gas sources already accounted for within our
inventory were removed to prevent redundancy.
Although our emissions map used for the model runs did not account for potential CH$_4$ emissions from wetland
sources, a series of wetlands emission scenarios was obtained for the region using data from Bloom, et al., (in review). From



this dataset, wetland $CH_4$ emissions make up only 1% of all regional $CH_4$ emissions in the most extreme scenario, and thus we
assume their impact is negligible to this study.

## 2.2 Aircraft Campaign

Observations for this project were obtained from a 3-week aircraft campaign during the period of May 14th-June 3rd, 2015.
The campaign was led by the Global Monitoring Division (GMD) of the National Oceanic and Atmospheric Administration
Earth Systems Research and Laboratory (NOAA ESRL), in collaboration with the University of Michigan. During this period,
the NOAA Twin Otter aircraft flew throughout the northeast portion of Pennsylvania, providing a total of ten flights across
nine days. The aircraft was equipped with a Cavity Ring-Down Spectroscopic analyser (Picarro G2401-m) measuring $CH_4$,
$CO_2$, CO, and water vapour mole fractions at approximately 0.5Hz with a random error of 1 ppb, 0.1 ppm, 4 ppb, and 50 ppm
respectively (Karion et al., 2013). GPS location, horizontal winds, temperature, humidity, and pressure were also recorded at
1Hz. The majority of observations for each flight occurred during the afternoon hours at heights generally lower than 1000 m
above ground level. Each flight contains at least one vertical profile within and above the boundary layer, with temperature
and water vapour observations from these profiles used to estimate the atmospheric boundary layer height and ensure that the
aircraft sampled air within the boundary layer throughout the flight. Observations taken above the boundary layer top are
flagged and removed from calculation.
Flight paths, wind speeds, and $CH_4$ observations for each of the 10 flights can be seen in Figure 3. For six of the ten
flights, a box pattern was flown around a large portion of unconventional natural gas wells in northeastern PA. These flights
were performed typically on days with a strong, steady wind, with a clearly defined upwind and downwind transect intended
for use in an aircraft mass balance calculation. Five of the six box-pattern flights were composed of two loops circling the gas
basin, allowing for two separate calculations of the upstream natural gas emission rate for the flight. On the remaining four
flights, raster patterns were performed to help identify spatial complexities of $CH_4$ emissions within the basin. All ten flights
were used in the model optimization calculation of the upstream natural gas emission rate.

## 2.3 Transport Model

The atmospheric transport model used in this study is the Advanced Weather Research and Forecasting (WRF) model (WRF-
ARW, Skamarock et al., 2008) version 3.6.1. The WRF configuration for the model physics used in this research includes the
use of: 1) the double-moment scheme (Thompson et al., 2004) for cloud microphysical processes, 2) the Kain-Fritsch scheme
(Kain and Fritsch 1990, Kain 2004) for cumulus parameterization on the 9-km grid, 3) the Rapid Radiative Transfer Method
for general circulation models (GCMs) (RRTMG, Mlawer et al., 1997, Iacono et al., 2008), 4) the level 2.5 TKE-predicting
MYNN planetary boundary layer (PBL) scheme (Nakanishi and Niino 2006), and 5) the Noah 4-layer land-surface model
(LSM) that predicts soil temperature and moisture (Chen and Dudhia 2001, Tewari et al., 2004) in addition to sensible and
latent heat fluxes between the land surface and atmosphere.



The WRF modelling system used for this study also has four-dimensional data assimilation (FDDA) capabilities to
allow meteorological observations to be assimilated into the model (Deng et al., 2009).  With WRF FDDA, observations are
assimilated through the entire simulation to ensure the optimal model solutions that combine both observation and the dynamic
solution, a technique referred to as dynamic analysis. Data assimilation can be accomplished by nudging the model solutions
toward gridded analyses based on observations (analysis nudging), or directly toward the individual observations (observation
nudging), with a multiscale grid-nesting assimilation framework typically using a combination of these two approaches (Deng
et al., 2009; Rogers et al., 2013).
The WRF model grid configuration used in this research contains two grids: 9- and 3-km (Figure 4), each with a mesh
of 202x202 grid points.  The 9-km grid contains the mid-Atlantic region, the entire northeastern United States east of Indiana,
parts of Canada, and a large area of the northern Atlantic Ocean. The 3-km grid contains the entire state of Pennsylvania and
most of the state of New York. Fifty vertical terrain-following model layers are used, with the centre point of the lowest model
layer located at ~10 m above ground level.  The thickness of the layers stays nearly constant with height within the lowest 1
kilometre, with 26 model layers below 850 hPa (~1550 m AGL).  One-way nesting is used so that information from the coarse
domain translates to the fine domain but no information from the fine domain translates to the coarse domain.
FDDA (Deng et al., 2009) was used in this research, with the same strategy as used in Rogers et al., (2013).  Both
analysis nudging and observation nudging were applied on the 9-km grid, and only observation nudging was applied on the 3-
km grid. In addition to assimilating observations and using the North America Regional Reanalysis model as initial conditions,
we reinitialize the WRF model every five days, allowing 12 hours of overlapping period in consideration of model spin-up
period to prevent model errors from growing over long periods. The observation data types assimilated include standard WMO
surface and upper-air observations distributed by the National Weather Service (NWS), available hourly for surface and 12-
hourly for upper air, and the Aircraft Communications Addressing and Reporting System (ACARS) commercial aircraft
observations, available anywhere in space and time with low-level observations near the major airports.
The WRF model used in this study enables the chemical transport option within the model allowing for the projection
of $CH_4$ concentrations throughout the domain. Surface $CH_4$ emissions used as input for the model come from our $CH_4$
emissions inventory and are all contained within the 3-km nested grid. Each source of $CH_4$ within our inventory is defined
with its own tracer (Table 1), allowing for the tracking of each individual source's contribution to the overall projected $CH_4$
enhancement within the model. For this study, $CH_4$ is treated as an inert gas. The potential for interaction with the hydroxyl
radical (OH), the main sink of $CH_4$, is neglected. A calculation assuming an above-average OH mole fraction over a rural
region of 0.5pptv (Stone et al., 2012) and a reaction rate of $6.5 \times 10^{-15}$ (Overend et al., 1975) produces a $CH_4$ sink of 0.5ppb per
hour. The duration of a flight can be up to 3 hours, leading to a potential loss of 1.5ppb over the course of a flight. This loss is
small but not insignificant. $CH_4$ plumes associated with natural gas during each flight ranged between 15-70 ppb, and a change
of 1.5ppb could theoretically impact observations by as much as 10% of the plume signal. However, this decrease in the $CH_4$
mole fraction would likely have equal impacts on both the background $CH_4$ values as well as the enhancement. Because
emission calculations are based on the relative difference between the $CH_4$ background mole fraction and the enhancement





downwind, it would take a gradient in the oxidation of OH to impact the results. Considering this relatively low destruction
rate, the expected homogeneity of the sink across the region, and the difficulties associated with the simulation of chemical
loss processes, we assumed that the CH$_4$ mass is conserved throughout the afternoon and therefore we ignored the impact of
oxidation by OH.
**2.4 Model Optimization Technique**
The objective of the model optimization technique is to solve for a natural gas emission rate as a percent of natural gas
production that creates the best match between modelled CH$_4$ concentration maps, provided by the transport model, with actual
CH$_4$ mole fraction observations, provided by the aircraft data. The optimization process in this study was originally designed
to solve for natural gas emission from unconventional wells and emissions from compressor facilities separately. Because the
flow rate of natural gas being processed was not available for each compressor station, emissions at each facility were originally
scaled based on the size of the station. However, when running the transport model using this emissions map, enhancements
from the compressor stations produced plume structures nearly identical in shape to enhancements from the unconventional
wells due to the similar spatial distributions of these two tracers. Without distinct differences between the enhancement patterns
from each tracer, it becomes impossible to distinguish which emissions source must be adjusted to obtain the closest match to
the observations. For this reason, emissions from compressor facilities are merged with unconventional well emissions in the
optimized emission rate. Though the emission rate solved for in this experiment only uses the locations and production for the
unconventional wells, this optimized rate represents emissions from both the wells and compressor facilities and will be
referred to as the modelled upstream natural gas emission rate. Midstream and downstream natural gas processes (such as
processing, transmission and distribution of the gas) and emissions from conventional wells are not solved for in this study
due to their minimal contribution (less than 5%) to CH$_4$ emissions in the region encompassed by the aircraft campaign.
Using the transport model WRF-Chem, CH$_4$ atmospheric enhancements were generated for each flight using different
tracers to track different components to the overall CH$_4$ enhancement (e.g. animal/animal waste, distribution sector, industries).
From these concentration fields, the upstream natural gas emission rate was solved for each flight using a three-step model
optimization technique. First, a background concentration was determined for each flight and subtracted from the observations
to create a set of "observed CH$_4$ enhancements," using
$$X_{EnhO} = X_{Obs} - X_{bg} \qquad , \qquad\qquad\qquad\qquad (1)$$
where $X_{Obs}$ is the CH$_4$ mole fraction observation from the aircraft, $X_{bg}$ is a chosen background value for the flight, and $X_{EnhO}$
is the calculated CH$_4$ enhancement at each observation. In this study, the background value is defined as the ambient CH$_4$ mole
fraction over the region not accounted for by any of the sources within the model, with each flight having a unique background
value. Box-pattern flights containing 2 loops around the basin may have a different background value assigned for each loop.
To determine the background mole fraction, we start with the value of the observed mole fraction in the lowest 2nd percentile
of all observations within the boundary layer for a given flight or loop. This chosen background value represents the CH$_4$ mole





fraction across the flight path from sources that are outside of our model domain. Because the background value is meant to
represent the $CH_4$ mole fraction outside the model domain which is otherwise unaccounted for in our model, using the
observations with the lowest $CH_4$ mole fraction is not always a sufficient definition for the background. On certain days, $CH_4$
enhancements from sources within the model domain can form plumes with wide spatial coverage that cover all observations
during a flight. For example, during a flight the lowest $CH_4$ observations from the aircraft may be 1850 ppb, but the model
simulation during that period indicates that all observations within the flight are being impacted by at minimum a 20 ppb
enhancement. In this case, we would set our background value for the flight at 1830 ppb, and say that our 1850 ppb observations
from the flight are a combination of an 1830 ppb background in addition to a 20 ppb enhancement from sources within the
model. By subtracting off this background value from our observations, we create a set of "observed $CH_4$ enhancements"
which can be directly compared to the model projected enhancements

The next step is to remove enhancements from this set that are not associated with emissions from upstream natural

gas using
$$X_{GasO} = X_{EnhO} - X_{OtherM} , \qquad\qquad (2)$$
where $X_{OtherM}$ is the modelled $CH_4$ enhancement at each observation from sources unrelated to upstream natural gas processes,
and $X_{GasO}$ is the observed $CH_4$ enhancement associated with upstream natural gas emissions for each observation. In this step,
each observed $CH_4$ enhancement has subtracted from it the projected non-natural gas enhancement from the model (i.e. nearest
grid point in space) using the corresponding model output time closest to the observation within a 20-minute time interval.
This creates a set of observed $CH_4$ enhancements related only to emissions from upstream gas processes, filtering out potential
signals from other $CH_4$ emitters and providing a set of observed enhancements that can be directly compared to the projected
upstream natural gas enhancement within the model. By subtracting these other sources from the observations, we make the
assumption that our emissions inventory is accurate for non-natural gas sources and that the transport of these emissions is
perfect, both of which are actually uncertain. Because errors exist in both the emissions and transport, it is possible to derive
a negative observed upstream gas enhancement if model-projected enhancements from other sources are larger than the
observed enhancement. To avoid such situation, these negative observed upstream gas enhancements are set to 0. Errors
associated with this issue and other uncertainties with our inventory are examined further in the results section of this paper.

In the final step, the upstream natural gas emission rate within the model is adjusted to create the best match between

the modelled upstream gas enhancement and observed upstream gas enhancement using
$$J = \sum_{i=1}^{n}(X_i^{GasO} - C * X_i^{GasM}) \qquad\qquad (3)$$
where $n$ is the number of observations in the flight, and $X_i^{GasO}$ and $X_i^{GasM}$ are the observed and modelled enhancement for
each observation $i$. In this equation, $J$ is a cost function we are trying to minimize by solving for a scalar multiplier $C$ which,
when applied to the modelled natural gas enhancements, creates the smallest sum of the differences between the observed
upstream gas enhancement and the modelled upstream enhancement. Because the emission rate within the model is linearly



proportional to the model enhancements, we can solve for the upstream natural gas emission rate that minimizes the cost
function using
$E = 0.13\,C$            (4)
where 0.13 was the first guess upstream emission rate (in percent of production) used in the model, and $E$ is the optimized
emission rate for the flight as a percentage of the natural gas production at each well. This final value represents an overall
emission rate associated with both unconventional wells and compressor stations across the region.
**2.5 Aircraft Mass Balance**
An aircraft mass balance calculation was performed for four applicable flights from the aircraft campaign as an alternative
method to calculate upstream natural gas emission rates independent of the transport model. The aircraft mass balance approach
uses the $CH_4$ enhancement between a downwind and upwind transect to calculate the total $CH_4$ flux of the area contained
between the two transects. We use the mass balance equation from Karion et al., (2013):
$E = \bar{U}cos(\bar{\theta}) \int_{-b}^{b} \Delta X \int_{z=0}^{z_{top}} n_{air} dz dx$            (5)
where $E$ is the total flux (in mol s$^{-1}$) coming from the enclosed flight track, $\bar{U}$ is the mean wind speed (in m s$^{-1}$), $\bar{\theta}$ is the mean
angle of the wind perpendicular to the flight track, $\Delta X$ is the $CH_4$ enhancement measured along the downwind flight track from
–b to b (expressed as a mole fraction), $n_{air}$ is the molar density of air within the boundary layer (in mol m$^{-3}$), and each of the
integrals represents the summing over all air being measured within our transect in both the horizontal (x) and the vertical (z).
By simplifying further and using the mean enhancement along each downwind transect as the enhancement and choosing $z_{top}$
to be the top of the boundary layer, we can transform the previous equation into the following:
$E = 37.3(L)(D)(\bar{U})\Delta\bar{X}cos(\bar{\theta})$            (6)
where L is the length of the transect (in meters), D is the depth of the boundary layer (in meters) found using observations
from vertical ascents during each flight, $\Delta\bar{X}$ is the mean enhancement across the transect (expressed as a mole fraction), $\bar{U}$ and
$\bar{\theta}$ are the mean wind speed (in m s$^{-1}$) and wind direction relative to the angle of the transect, and 37.3 is the average molar
density of dry air within the boundary layer (in mol m$^{-3}$) assuming an average temperature and pressure of 290K and 900hPa.
Flights on May 22nd, May 23rd, May 28th, and May 29[th], 2015 were selected for mass balance calculations based on
their box-shaped flight patterns surrounding a large portion of the northeastern Marcellus gas basin, reasonable steady-state
wind conditions during the flight (this eliminated May 14[th] when a low pressure system was stationed over the region), and a
lack of strong $CH_4$ enhancements originating from outside of the box which could affect both the upwind and downwind
transects. Emission rates within the enclosed region were calculated from each of these flights, and emissions not associated
with unconventional gas production and gathering sources within the box were subtracted out using information from our
inventory. A ratio was taken of the calculated emission rate to the total production in the box, and an upstream natural gas



emission rate based on production was obtained. For each of the May 22nd, May 23rd, and May 28th flights, two loops of
similar structure and at similar altitudes were performed around the basin, allowing for two individual mass balance
calculations on these days. An emission rate is calculated for each of these loops and the two values are averaged to calculate
the daily emission rate.

## 364 3 Results

### 365 3.1 Methane Inventory

From the $CH_4$ inventory created in this study, a total anthropogenic $CH_4$ emission rate of 2.76 Tg $CH_4$ year$^{-1}$ is projected within
our inner model domain (Figure 5) with values for individual source contributions shown in Table 2. This total emissions
estimate assumes a leak rate of 0.13% of gas production for unconventional wells, and does not account for emissions from
natural gas transmission and storage facilities outside of PA due to a lack of information available from other states. Within
the model domain, the area encompassing southwestern PA and northeastern WV stands out as the largest contributor to $CH_4$
emissions, with emissions from conventional gas, unconventional gas, and coal mines all having significant contributions to
the total. In particular, the large emissions from coal make this region unique in comparison to other shales. The EPA's
Greenhouse Gas Reporting Program dataset for the year 2014 lists individual coal mines in the southwestern portion of our
domain as 8 of the top 10 $CH_4$ emitting facilities across the entire United States. This large area source of $CH_4$ can have an
impact on $CH_4$ concentrations hundreds of kilometres downwind and must be taken into account when winds are from the
southwest (Figure 6). Examples of this plume and its impacts on the aircraft campaign are discussed in Section 3.2.1.

### 378 3.2 Model Optimization Results

### 379 3.2.1 Case Studies

From the aircraft campaign, a total of 10 flights across 9 days were used in the model optimization technique. For each one of
these flights, $CH_4$ concentration fields were produced using WRF-Chem, and the emission rate from upstream gas processes
was adjusted as outlined in the methods section to find the rate that best matches the total observed $CH_4$ enhancement. For box
flights with two loops completed around the basin, emission rates were calculated for each loop independent from one another
and then averaged for the flight. Table 3 provides the general meteorology for the 10 flights.
During each of the observational periods, we use the transport model to project the mole fraction enhancement across
the region for each of the different $CH_4$ tracers (Figure 7). From these projections, we see three common sources of $CH_4$ which
can significantly influence the observed mole fractions in our study region of northeastern PA. The first is emissions from
unconventional gas in northeastern PA. Although the first-guess total emissions from upstream production in the Marcellus
are small compared to the overall contributions from other sources within the domain, their proximity to the aircraft track



results in unconventional gas having the largest contribution to observed enhancements throughout the domain covered by
most of the flights, often producing signals downwind of about 20-80 ppb above background levels. The second most
influential source of enhancements in our study region comes from various sources of $CH_4$ emissions located in southwestern
PA. Despite being more than 400 km away from our study region, large plumes from coal and other sources in the southwestern
corner of the state can contribute enhancements as high as 50 ppb across portions of the flight when winds are from the
southwest, affecting background measurements and masking signals from the unconventional gas. One final, but less
influential source of $CH_4$ enhancement is animal agriculture in southeastern PA. Lancaster County is home to roughly 20% of
all cattle in the state, with more than 200,000 cattle and calves as of 2012. A southerly wind can result in a 5-15 ppb
enhancement across the flight path due to enteric fermentation and manure management from these cattle. Because of coal,
conventional gas, and cattle sources located south of the basin, signals from flights with a southerly component to the wind
can be difficult to interpret without modelling the projected plumes associated with these sources. Observations on these days
contrast to days with a northerly wind component, where a lack of $CH_4$ sources north of the study region results in observations
with a more clearly defined background and unconventional natural gas enhancement.

For each of the ten flights, variability in the model-observation offset was observed. The first loop of the May 29th

flight is the best example of a case where comparisons between the modelled and observed enhancements match closely after
optimization. For this flight, a box pattern was flown encompassing a majority of the unconventional wells in northeastern PA,
and enhancements were observed along the western and northern transects of the flight. Modelled enhancements from sources
unrelated to upstream gas emissions showed a broad $CH_4$ plume associated mostly with animal agriculture along the western
edge of the flight, and a smaller enhancement on the eastern edge associated with two landfills in the Scranton/Wilkes-Barre
urban corridor (Figure 8). Both of these enhancements are subtracted off from the observations to produce a set of observed
enhancements due to natural gas production and gathering facilities. Any enhancements in this new observational dataset are
located almost entirely along the northern transect of the flight, directly downwind of the natural gas activity in the region.
The observed upstream gas enhancement is then directly compared to the modelled upstream enhancement using its first guess
emission rate, and an optimized upstream emission rate of 0.26% of production (i.e. a doubling of the first guess) is calculated
by minimizing the difference between the two datasets (Figure 9).

The match between observed and modelled $CH_4$ enhancements on the first loop of the May 29[th] flight is closer than

any other flight in the campaign, with a correlation coefficient of 0.73. The success of the model on this day is likely due to a
number of ideal conditions. In general, inconsistencies between the modelled and observed mean wind speeds and boundary
layer heights can have a linear bias on the projected enhancements, but for this flight differences between the observed and
modelled wind speed and boundary layer height were near 0 for both loops (Figure 10, 11). Observed wind directions
throughout the course of the flight had little spread, resulting in a transport of the $CH_4$ plumes that the model was able to match
well. Furthermore, the observed mean wind speed was 4.6 m s$^{-1}$, a moderate wind which allows for a steady transport of any
enhancements towards the downwind transect, but not strong enough to dilute their magnitude, resulting in an easily observable
enhancement downwind of the basin. Finally, intrusions from sources unrelated to upstream gas were small on this day due to

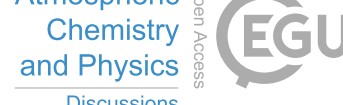



favourable wind conditions, reducing the probability of incorrectly attributing the observed enhancements to the wrong source.
Enhancements from upstream natural gas processes were around 30 ppb along our downwind transect. By comparison,
enhancements from other sources were lower than 15 ppb along a majority of the flight, and most of these enhancements were
located west of the downwind transect, making them easier to identify and remove without unintentionally impacting
enhancements from the natural gas plume. All of these different factors likely contributed to producing a situation where the
model was successfully able to match $CH_4$ observations during the May 29th flight.

Flights that occurred on days with a southwest wind had a tendency to produce $CH_4$ observations that were intuitively

difficult to interpret due to convolved $CH_4$ sources in southwestern Pennsylvania. One of these complex observation sets
occurred during the late afternoon flight on May 24th, 2015 (Figure 12). Observations on this day show a $CH_4$ enhancement
pattern that decreased with latitude, with higher $CH_4$ mole fractions observed farther south. Given the location of the wells in
the middle of the flight path and the WSW wind pattern in the region, this north/south $CH_4$ gradient is unexpected and
counterintuitive compared to where one would expect the enhancements to be based solely on the presence of the gas industry
in northeastern PA. However, through modelling each of the many contributors of $CH_4$ within our inventory, we are able to
recreate this latitudinal $CH_4$ gradient and better understand the observed patterns (Figure 12). Throughout an 18-hour period
leading up to the May 24th flight, winds from the SSW transport emissions from coal in southwestern PA northeastward until
they reach the centre of the state, where a westerly wind then shifts the plume across the study region such that it only intersects
the southern half of the flight path. Because of both the magnitude of the coal emissions and an accumulation that occurred in
the southwestern portion of the state during the previous night, the modelled enhancement from the coal plume is substantial
(>20 ppb) as it crosses over the flight path and covers up much of the signal from upstream gas emissions. Nonetheless, the
transport model is able to account for these far-reaching sources and separate out their contribution to the observed
enhancements. We are able to solve for the optimal upstream natural gas emission rate and recreate the May 24th flight
observations more accurately than most other flights, with a correlation coefficient of 0.71 between the observations and model
values.

Despite the model's success at recreating observations from the May 24th late-afternoon flight, there is reason to be

careful when interpreting results from observations influenced by distant sources. In particular, some transport error is
unavoidable in atmospheric reanalyses, and the longer the time and distance a plume takes to reach the observations, the more
its position and magnitude will be susceptible to these errors. During the early May 24th flight, a small 50 km shift in the
location of the coal plume across the study region would change projected enhancements at some observations by as much as
20 ppb. Furthermore, errors in the transport speed could create scenarios where the coal plume either arrives in the study region
too early or exits too late, creating a projected enhancement pattern that does not agree with the observations (Figure 13).
Additionally, inaccuracies with the emission estimates of non-unconventional gas sources in the inventory will impact the
magnitude of their $CH_4$ enhancements, creating additional errors in the optimization process when subtracting out these
enhancements from the observations. The early-afternoon May 24th flight and May 25th flight are both examples where
influences from $CH_4$ sources in southwest PA create complex structures in the enhancements, which the model is not able to



match as well as the late-afternoon flight on May 24[th] (Figure 14). And although observations and modelled enhancements
closely match throughout portions of these two flights, a slight shift in the modelled wind direction can lead to vastly differing
results due to the large offset small changes in the wind field can have on an emission source hundreds of kilometres away.
Thus, results from the flights on May 24[th] and May 25[th] should be taken with caution. A deeper analysis of these errors can be
found in Section 3.2.2.
Though transport errors from far off sources can have significant impacts on the optimized emission rate, more local
transport error is handled more effectively in the model through use of the error minimization cost function discussed in the
methods section (Eq. 3). Through use of this cost function, we solve for an upstream emission rate that minimizes the area
under the curve between the observed and modelled enhancement. This analysis produces a compensation effect that can adjust
for misalignment between the observed natural gas $CH_4$ plume and the modelled plume, and works best for local sources
whose plume structure is similar between the model and observations but location is misaligned. The impact of this effect can
be seen best for the flight of May 14[th], 2015 (Figure 15). During this day, high pressure was centred just north of the flight
track, creating diverging winds that were difficult for the transport model to simulate, and the major plume associated with
upstream gas emissions ends up south of a well-defined plume in the observations. Despite this misalignment, the model still
simulates the correct magnitude and width of the plume, providing confidence that small errors in the transport of local, well-
defined structures have only a small impact on the overall emission rate calculation for any given flight. This assumption is
corroborated by an analysis of simulated enhancements of an aircraft transecting a plume at an angle of 45° with errors in the
projected wind direction varying between ±20°. Such angle would produce errors in the optimized emission rate ranging from
±30% (Figure 16). These errors become smaller as the angle of the aircraft's path with the axis of the major plume approach
90°. Because the downwind transect of most flights is often close to perpendicular with the plume from upstream gas sources,
and differences between modelled and observed winds are often less than 20°, emission estimate errors associated with
incorrect transport of the $CH_4$ plumes would be less than 30%.
**3.2.2 Emission Rates and Uncertainty Assessment**
For each of the ten flights, an uncertainty assessment was performed to obtain a range of likely upstream emission rates for
any individual flight. Five different sources of error were considered in this assessment: model wind speed error, model
boundary layer height error, $CH_4$ background error, $CH_4$ emission inventory error, and model/observation mismatch error.
These five sources of error vary substantially from flight to flight depending on conditions, and each can have significant
impacts on the total uncertainty.
Errors in the modelled wind direction and boundary layer height have impacts on our emission estimates which have
a linear impact on the results. If we assume a constant wind speed, a constant boundary layer height, and no entrainment of air
from the top of the boundary layer, we use the following equation to understand these impacts.
$\Delta C = \overline{F_0}(\frac{\Delta x}{U*D})$ (5)




where $\Delta C$ is the total $CH_4$ enhancement of the column of air contained within the boundary layer, $\overline{F_0}$ is the average emission
rate over the path the parcel travelled, $\Delta x$ is the distance the column of air travelled, $U$ is the wind speed and $D$ is the boundary
layer height. Using this equation, we can see the linear relationship between the model wind speed, model boundary layer
height, and the calculated emission rate. As an example, if wind speeds in the model are biased low, natural gas enhancements
projected by the model would increase inversely. To compensate for this effect, the optimized emission rate would decrease
proportionally. A similar case can be made for bias in the boundary layer height. Both errors in the wind speed and boundary
layer height have known impacts on the optimized emission rate which can be corrected for, so long as the errors of each are
known.

To calculate the error in the model wind speed, we assume aircraft observations are truth and use

$U_e = \frac{\overline{U}_m - \overline{U}_{obs}}{\overline{U}_{obs}}$                    (6)
where $\overline{U}_{obs}$ is the mean observed wind speed by the aircraft across all points within the boundary layer, $\overline{U}_m$ is the mean
modelled wind speed by the model across all points closest in time and space to each observation, and $U_e$ is the wind speed
error percentage.

To compute the error in the modelled boundary layer height, the observed boundary layer height for each flight is

assumed to be the true boundary layer height and the boundary layer height percentage error, $H_e$, is estimated using:
$H_e = \frac{\overline{H}_m - \overline{H}_{obs}}{\overline{H}_{obs}}$                    (7)
where $\overline{H}_{obs}$ is the average observed boundary layer height across each of the aircraft profiles for a given flight, $\overline{H}_m$ is the model
boundary layer height closest in time and space to the location of the observed profiles averaged over all profiles. For both the
observation and the model, boundary layer heights were determined by locating height of the potential temperature inversion
associated with the top of the boundary layer (Figure 11). In cases where a potential temperature inversion could not easily be
identified, changes in water vapour, $CO_2$ and $CH_4$ mixing ratios were used to identify the boundary layer top.

Errors in the model wind speed and boundary layer height are calculated for each of the ten flights. From these errors,

a corrected optimized emission rate is calculated for each flight using Eq. (8):
$E_{new} = \frac{E}{(1+U_e)(1+H_e)}$                    (8)
where $E$ is the original emission rate and $E_{new}$ is the corrected optimized upstream natural gas emission rate as a percent of
production. Table 4 shows the wind speed and boundary layer height errors for each flight as well as the optimized and
corrected natural gas emission rates. On days where model performance was poor in regards to the wind speed and boundary
layer height, we can see changes in the corrected emission rate. For most days, this change is less than 20% different than the
original optimized emission rate. However, both May 14th and May 25th have corrected emission rates which are around a
factor of 2 different from their original value. Whether these corrected emission rates are more accurate than the original





optimized rates is debatable. To calculate these alternative emission rates, we must assume that the wind speeds and boundary
layer heights from our limited number of observations are the true values in the atmosphere, which may not be the case.
Regardless of which rate is more accurate for each flight, the overall 16% high bias in the model wind speed and the -12% low
bias in the model boundary layer result in compensating errors that cancel out, and the mean emission rates across all flights
end up equal. Thus, any errors associated with these two meteorological variables has a trivial impact on the overall calculated
emission rate for the region and the uncorrected emission rates are used for the final mean and uncertainty calculations.

In addition to errors related to wind speed and boundary layer height, we quantify three other sources of error in each

flight: errors in the selected $CH_4$ background value, errors in the $CH_4$ inventory, and errors associated with the overall model
performance (Table 5). Unlike the wind speed and boundary layer errors which have easily computable impacts on the emission
estimates, these other three sources of error and their impact on the optimized emission rate are more difficult to quantify.

The background error relates to the value chosen for each flight which represents the ambient $CH_4$ concentration in

the boundary layer unrelated to emission sources within the model. In this study background values ranged from 1897-
1923ppb. Though background values should not have high variability during a 2-3 hour mid-afternoon flight, entrainment from
the boundary layer top can lead to the mixing in of tropospheric air that has different $CH_4$ mole fraction values from those
within the boundary layer, resulting in a change in the afternoon background value with time. Furthermore, for days on which
all aircraft observations (including those upwind of the unconventional wells) are impacted by various $CH_4$ plumes predicted
within the model, it is difficult to determine the background $CH_4$ concentration accurately. Additionally, observations
corresponding to locations with no modelled enhancement may in fact have been impacted by missing sources in our inventory,
highlighting the difficult nature of knowing with certainty where and what the background is for any given flight.
Understanding this uncertainty is crucial; any error in subtracting off the background value directly impacts each observation's
observed natural gas enhancement. For example, a background value of 1 ppb below the true background for a given flight
would add 1 ppb to each observed natural gas enhancement for all observations, creating a high bias with the optimized
upstream emission rate. To account for this error, each flight's optimization processes was rerun iterating the background value
by ±5 ppb, and the ratio of the percent change in the emission rate compared to the original case was defined as the resulting
error in the emission rate due to background uncertainty. This ±5 ppb background error range is an estimate at the range of
possible error in the background based on changes observed in the upwind measurements from each of the flights and is meant
to be a conservative estimate of the error. The impact this error can have in the emission rate varies depending on the magnitude
of the observed downwind enhancements during a flight. A plume containing a $CH_4$ enhancement of 50 ppb will have a smaller
relative error from a 5 ppb change compared to one with an enhancement of only 10 ppb. Thus, days with high wind speeds
and a high boundary layer height (and thus enhancements of a smaller magnitude) tend to be affected the most by background
errors.

Similar to background errors, errors from the $CH_4$ emissions inventory are difficult to quantify. In the model

optimization technique, we subtract out enhancements from sources unrelated to unconventional natural gas before solving for
the upstream gas emission rate. In doing so, we are making the assumption that our emissions inventory for sources unrelated



to upstream natural gas processes are accurate. In truth, each emission source in our inventory comes from a different dataset
and has its own unique error bounds, many of which are unknown. Because of the potential for errors in these emission
estimates, we take a conservative approach and iterate the unconventional emissions optimization approach for each flight,
varying the emissions from other unsolved for emission sources in the model by a factor of 2 in each direction, thus applying
a range of 50-200% to the emissions inventory values to assess its impact on the calculated upstream natural gas emission rate.
Despite the extensive range of emissions used in the error analysis, its impact is minimal on most days due to the northeastern
Marcellus region having very few emission sources not related to upstream gas processes. Only for the flights on May 24th do
we see errors from the inventory contribute significantly to the overall daily error, when the coal plume in southwestern PA
enters the centre of the study region and has a large role in the upstream emission rate calculation for that day.

The final source of error accounted for attempts to quantify the similarity of the pattern of modelled and observed

natural gas enhancements, referred to here as the model performance error. Figure 17 shows an example of two days, one of
which the model appears to recreate the observations, and the other of which the model poorly matches the shape of the
observed enhancements. Comparing these two simulations with no other information, we hypothesize that one should put more
trust in the upstream natural gas emission rate calculated for the flight whose modelled upstream enhancements match
structurally compared to the emission rate from the flight whose modelled enhancement bares little semblance to the observed
enhancement. The model performance error is designed to account for the trustworthiness of the optimized upstream emission
rate based on how well the model simulates a given day. The model performance error is calculated using Eq. (9):
$e_{Perf} = \frac{\bar{\sigma}_{\Delta X}}{\Delta X_{gas}}$            (9)
In this equation, $\bar{\sigma}_{\Delta X}$ is the standard deviation of the difference between the modelled and observed upstream natural gas $CH_4$
enhancement using the optimized emission rate, and $\Delta X_{gas}$ is the observed magnitude of enhancement from the major natural
gas plume observed in each flight. Here, $\Delta X_{gas}$ serves as a normalization factor to account for the varying strength of the
enhancement from flight to flight, and ensures that days with increased enhancements due to meteorological conditions or true
daily fluctuations in the upstream natural gas emissions do not proportionally impact the performance error percentage. For
example, a day with high winds and a deep boundary layer would produce smaller enhancements, leading to a small $\bar{\sigma}_{\Delta X}$
regardless of model performance unless normalized by $\Delta X_{gas}$.

Table 5 summarizes the background error, inventory error, and model performance error, and assumes independence

between the three error sources to calculate the total error for each flight. The largest error occurred during the May 22nd flight,
where an unexplained enhancement along the northern transect led to a poor match between the modelled enhancements and
the observed enhancements. This may explain the anomalously high optimized emission rate for that day. Other flights with
large error are those which occurred on May 24th and May 25th, both days where enhancements from southwestern PA are
believed to be influencing observations.



Based on the conservative methodology used to calculate these errors, we assume the total error for each flight
represents a 2σ range of possible emission rates and calculate a weighted mean and a 2σ confidence interval for the overall
upstream emission rate across the ten flights. From this approach, we find a mean upstream emission rate of 0.36% of
production and a 2σ confidence interval from 0.27-0.45% of production.
**3.3 Aircraft Mass Balance Results**
In addition to the model emission optimization, a simplified aircraft mass balance technique was used to calculate upstream
natural gas emission rates for flights with a box-pattern, a consistent wind direction within the box, and minimal intrusion of
$CH_4$ enhancements from outside the study region that would affect both the upwind and downwind transects. Of the 6 days
with box-patterned flights from the aircraft campaign, one day (May 14th) contained a surface low-pressure centre in the middle
of the flight resulting in erratic wind patterns, and another day (May 25th) had $CH_4$ plumes from southwestern PA affecting
portions the flight observations. These days were not used for a mass balance, and calculations were performed for the
remaining box-pattern flights (May 22nd, May 23rd, May 28th, May 29th). Of the four remaining flights, three of these flights
contained two loops around a portion of the Marcellus basin. A mass balance was performed on each loop, resulting in a total
of 7 mass balance calculations for the region across 4 days. Table 6 summarizes the results from the mass balance flights.
For each flight, a total flux within the box encompassed was calculated using Eq. (6). Using this flux, a natural gas
emission rate based on production from within the box was calculated using Eq. (10)
$E_\% = \frac{E - E_{other}}{P}$ (10)
where $E$ is the total flux from Eq. (6) (in kg hr$^{-1}$), $E_{other}$ are the emissions enclosed in the box from sources not related to
upstream natural gas processes (in kg hr$^{-1}$), $P$ is the total $CH_4$ from natural gas being produced within the box (in kg hr$^{-1}$), and
$E_\%$ is the resulting natural gas emission rate as a percent of total production within the box. Calculated emission rates varied
extensively between flights, ranging from 0.11% to 1.04% of natural gas production. Comparing emission rates between loops
on the same day, we see more consistency in the values. This result is not surprising, as on each of the days with multiple
loops, upwind and downwind $CH_4$ concentrations patterns tended to be similar between loops. Thus, differences in the total
emission rate are likely due to either errors specific to each day (such as background variability, errors in meteorology) or real
daily variability in the upstream natural gas emission rate.
As an error analysis for the mass balance flight, we look at four potential sources of error (Table 7). One source of
uncertainty comes from the observed wind speed used in Eq. (6). For our experiment, we take the mean observed wind speed
from the aircraft and assume this value represents the mean wind speed within the entire box during the 2-3 hour period it
would take for air to travel from the upwind transect to the downwind transect. To understand the uncertainty and biases
associated with this assumption, we recreate wind observations along the flight path using values from WRF-Chem, and
compare the mean wind speed from the simulated observations to the mean model winds contained within the box integrated
throughout the boundary layer during the 3 hour period closest to the flight time. By making this comparison, we are able to





understand the representation error associated with treating the wind speed observations from the aircraft as the wind speed within the entire box during the period it would take for air to cross from the upwind transect to the downwind transect. On average, modelled wind speeds following the flight were 7% faster than integrated wind speeds within the box, due to the inability for aircraft observations to account for slower wind speeds closer to the surface. This bias was removed from each day's calculated wind speed. After accounting for the wind speed bias, the average error of the modelled wind speed following the flight path compared to the modelled winds within the box was 3%. This 3% uncertainty was applied to each flight and used as the potential uncertainty in the mean wind speed. Errors in the wind direction were neglected, as each flight used in the mass balance completely surrounded the basin using downwind transects at multiple angles, and thus small errors in the wind angle would result in a negligible net change on the total flux calculated.

Another source of uncertainty is error in the boundary layer height. For each flight, between 2-3 vertical profiles were performed, and the mean height was used in Eq. (6). The standard deviation of different heights from each transect was used as the uncertainty. On May 22$^{nd}$, a boundary layer height could be interpreted from only one vertical transect. For this day, we assume an uncertainty of ±200m (±9%).

Uncertainty in the CH$_4$ background mole fraction was estimated similarly to the boundary layer height. On three of the four flights, two upwind transects were performed. The mean observed CH$_4$ mole fraction between the two transects was used as the background value for the entire flight, and the standard deviation between the loops was used as the uncertainty. On both the May 23$^{rd}$ and May 28$^{th}$ flights, background differences between the two transects were less than the instrument error of 1 ppb. On these days, we use the instrument error as the background error. On May 22$^{nd}$, only one upwind transect was usable for the calculation. For this day, we assume a conservative estimate in the uncertainty of the background of ±5 ppb.

Finally, we assess uncertainty in the emissions inventory. After a CH$_4$ flux is calculated for each loop, emissions from sources contained within the box that are not associated with upstream natural gas processes must be subtracted out to solve for the upstream natural gas emission rate. Any errors associated with our inventory will result in a CH$_4$ source attribution error. To account for the potentially large uncertainty with the emission sources in our inventory, we vary these non-natural gas emissions by a factor of 2 to test the impact on the solved upstream natural gas emission rate. Because northeastern Pennsylvania contains few sources of CH$_4$ emissions outside of natural gas production, the impact of this uncertainty is typically less than 20% of the total emissions calculated within the box.

From Table 7, we can see the largest relative errors occur on the May 22$^{nd}$ flight. It is on this day where we have the largest uncertainty in the background value, with observations towards the end of the flight becoming unusable due to a rapid and unexplained decrease in the CH$_4$ mole fraction with time (Figure 18). This day also features the highest boundary layer height and fastest winds of all flights done in this study, reducing the magnitude of the enhancement associated with the natural gas plume and thus amplifying the effects an uncertain background has on the overall uncertainty of the calculated CH$_4$ flux. Uncertainty across the other three flights is smaller, and results between individual loops on the May 23$^{rd}$ and May 28$^{th}$ flight provide more confidence in the calculated flux for those days.



Using the mean estimated CH$_4$ emissions and uncertainty for each loop, we calculate a daily mean emission rate and
uncertainty for each of the four days. We then solve for an unweighted mean across the four flights to derive our overall
emissions estimate from the aircraft mass balance approach, and use the standard error of the flights to estimate the uncertainty.
In doing so, we derive a natural gas emission rate from upstream processes of 0.34% of production, with a 2σ confidence
interval from 0.06-0.62% of production. Here, we use the arithmetic mean rather than a weighted mean due to the linear
relationship between the size of the emission rate and the size of the errors. Because errors associated with ABL height and
wind speed have a proportional impact on the calculated emissions within the box, days with a high emissions estimate produce
large uncertainties relative to days with a small emission rate. Using a weighted mean approach assigns more weight to the
days with low estimated emissions, and produces an overall emission estimate too low and certain to have confidence in
(0.12±0.02 percent of gas production).

## 4 Discussion

### 4.1 Upstream Emission Rate

From this study, we estimate with a 2σ confidence interval an emission rate between 0.27-0.45% of gas production using the
model optimization method and 0.06-0.62% of gas production using the aircraft mass balance. Figure 19 provides the emission
range estimates from upstream natural gas processes using both the model optimization technique and mass balance technique
when applicable. These emission rate estimates as a percent of production are the lowest observed from top-down
measurements of different basins in the U.S., and raise questions as to why these values in the northeastern Marcellus region
appear to be low. One possibility may be related to the well efficiency of the northeastern Marcellus region compared to other
major shale plays (Table 8). In terms of gas production per unconventional well, the Marcellus is the highest of all major basins
in the U.S. Furthermore, the gas production per well increases by nearly a factor of two when focusing specifically on
Susquehanna and Bradford Counties in northeastern Pennsylvania where the majority of the wells from this study are located
(Figure 20). The large difference in production per well between the northeastern Marcellus and other shales may partly explain
the low emission rates as a percentage of production. Throughout this study, we normalize natural gas emissions as a percentage
of total production under the assumption that higher throughput of natural gas in a system should lead to higher emissions in
the system. However, if leaks are more influenced by the number of pneumatic devices rather than the throughput passing
through the device, a high-efficiency system such as the northeastern Marcellus could end up having a very low emission rate
as a percentage of production, but a similar emission rate compared to other basins based on the number of wells, compressors,
etc. A thorough bottom-up study of the Marcellus region measuring emissions on a device level could provide an answer to
this hypothesis.



## 4.2 Advantages of Combining Observations with Model Output

One of the major advantages of using a chemical transport model to solve for natural gas emission rates compared to a standard mass-balance approach is that the transport model is able to account for the complex and oftentimes non-uniform plume structures originating from sources outside the flight path that can affect observations. When performing a mass balance over a basin, it is assumed that the upwind transect is representative of the air exiting the downwind transect after subtracting out all sources within the box. However, this assumption is only true if winds contained within the flight path are in perfect steady state during the time it take for air to move from the upwind transect to the downwind transect, and that measurements from the downwind transect occurred at a much later time so that the air being measured is the same air measured from the upwind transect. These conditions are not easily achieved for regional scale mass balances due to the long times needed for the air from the upwind transect to reach the downwind transect. As an example, from the four mass balance flights performed for this study the average time for air to move from the upwind transect to the downwind transect was 4 hours whereas the average time between the aircraft's upwind and downwind measurements was ~40 minutes. The aircraft observations can be thought of as a snapshot in time, which can be problematic if large scale plumes from outside the domain are moving through the region and impacting only certain portions of the observations during the flight's short timeframe. By using a transport model for a domain much larger than that of the flight paths, we are able to track these far-reaching plumes and identify situations where the background $CH_4$ concentrations may be spatially heterogeneous.

The potential usefulness of using a transport model alongside a mass balance calculation can best be demonstrated from observations taken over the Marcellus during a 2013 aircraft campaign (Peischl et. al 2015). During this flight the prevailing winds were from the WSW, and the largest $CH_4$ enhancements were observed along the western edge of the flight path, upwind of the unconventional wells. Using our transport model, we are able to recreate the day of flight and attempt to use our inventory and explain this feature (Figure 21). Comparisons between modelled output and observations show a 60 ppb $CH_4$ enhancement from coal and conventional wells in southwest PA stretching close to the western edge of the aircraft observations, a plume structure similar to the one observed during the May 24th flight from our own study. Though this plume does not initially align with the observed transect with the largest enhancements, we recognize that the coal and gas plume travels for more than 20 hours (a distance of 400 km) from its source before reaching the flight path. If we allow for a 10% error in the transport speed and therefore advance the transport model by an additional two hours past the time in which the aircraft observed these high values, we are able to line up the centre of the plume with the largest observed $CH_4$ mole fractions along the western edge of the flight. In addition to the 60 ppb enhancement along the centre of the plume, the model projects 20 ppb enhancements along the edges and in front of the plume centre. These smaller enhancements have an influence along different portions of the flight which varies in magnitude, making it difficult to assess a proper background $CH_4$ value upwind of the wells and potentially masking natural gas enhancements downwind of them. But by using a transport model, we are able to see the potential impact of these far-reaching sources which would otherwise not be considered in a regional mass balance and better understand the complex $CH_4$ plume structures which can occur in a given region under specific wind conditions.



## 5 Conclusion

Using the model optimization technique presented in this study, we find a weighted mean natural gas emission rate from unconventional production and gathering facilities of 0.36% of production with a 2σ confidence interval from 0.27-0.45% of production. This emission rate is supported by four mass balance calculations, which produce a mean of 0.34% and a 2σ confidence interval from of 0.06-0.62% of production. Applied to all the wells in our study region, this mean rate results in a leakage rate of 20 Mg $CH_4$ $hr^{-1}$. The emission rate found in this top-down study quantified as a percent of production is significantly lower than rates found using top-down methodology at any other basin, and indicates the presence of some fundamental difference in the northeastern Marcellus gas industry that is resulting in more efficient extraction and processing of the natural gas.

The ten flights that took place in this study reveal large regional variations in the $CH_4$ enhancement patterns depending on the prevailing wind direction. On days with a northwest wind, observed enhancements come primarily from the natural gas industry, and a small plume associated with it can be seen on the downwind leg of each flight with few enhancements upwind of the wells. Flights which took place with winds conditions predominantly from the southwest were more difficult to interpret. Plumes associated with coal and other potential sources of $CH_4$ in the southwestern Pennsylvania create complex enhancement patterns affecting both the upwind and downwind portions of the flight, making both the background $CH_4$ mole fraction and enhancements from the gas industry difficult to interpret. The stark difference between observations which occurred with a northwest wind compared to a southwest wind illustrates the importance of having multiple flights across days with various wind conditions to better understand the major influences on $CH_4$ concentrations throughout a region. The regional influences in Pennsylvania also demonstrate the utility of deriving an emissions inventory that provides input data to drive a transport model, allowing one to forecast $CH_4$ mole fractions on difficult days and better understand the daily uncertainties associated with heterogeneous background conditions.

Though this study presented observations from ten flights over a three-week period, it is not able to account for the potential of long term temporal variability in the emission rates. In May 2015 when the flights took place, the entire Marcellus basin was nearing peak production and active drilling and hydraulic fracturing was still ongoing in the region. By mid-2016, the rate of drilling of new wells in the northeast Marcellus had decreased and natural gas production had begun to decline in the area. A snapshot of the emission rate during one month of a basin in its peak production is insufficient to characterize emissions from an area that is likely to be producing and transporting gas at various intensities for decades. We need to quantify the long-term climatological impacts of gas production. Future work examining the temporal variability of $CH_4$ emissions within natural gas basins would complement short-term, high-intensity studies such as this one, and aid with understanding how well the calculated emission rates represent the gas basin over the course of time.



## Acknowledgements

This work has been funded by the U.S. Department of Energy National Energy Technology Laboratory (project DE-
FE0013590). We thank in-kind contributions from the Global Monitoring Division of the National Oceanic and Atmospheric
Administration, and from the Earth and Environmental Systems Institute, the Department of Meteorology and Atmospheric
Science, and the College of Earth and Mineral Science of The Pennsylvania State University. We also want to thank the
Pennsylvania College of Technology in Williamsport, PA for access to their Technology Aviation Center facilities during the
aircraft campaign. We also want to thank Lillie Langlois from the Department of Ecosystem Science and Management for
sharing pipeline information, Anthony J. Marchese and Dan Zimmerle (Colorado State University) for information on
compressor stations, Jeff Peischl for sharing data from his 2013 flight campaign, and Bernd Haupt (Penn State University) for
data processing and management during the project. Finally, we would like to thank Dennis and Joan Thomson for their
creation and continued support of the Thomson Distinguished Graduate Fellowship.



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



**Table 1: List of tracers used in the transport model.**

| Tracer # | Name | Description of source |
|---|---|---|
| 1 | Unconventional Wells | Emissions from unconventional wells. |
| 2 | Storage Facilities | Emissions from compressors associated with natural gas storage. |
| 3 | Pipelines | Emissions from gathering and transmission pipelines |
| 4 | Distribution | Emissions from the distribution sector of the natural gas industry. |
| 5 | Conventional Wells | Emissions from conventional wells. |
| 6 | Landfills/Other | Emissions from landfills and uncharacterized industrial sources. |
| 7 | Coal | Emissions from active and abandoned coal mining. |
| 8 | Animals/Waste | Emissions from enteric fermentation and manure management |
| 9 | Production Compressors (HP) | Emissions from compressor stations characterized as "production". Emissions scaled linearly with wattage. |
| 10 | Gathering Compressors (HP) | Emissions from compressor stations characterized as "gathering". Emissions scaled linearly with wattage. |
| 11 | Other Compressors (HP) | Emissions from all other compressor stations. Emissions scaled linearly with wattage. |
| 12 | Production Compressors (C) | Emissions from compressor stations characterized as "production". Emissions constant among compressors. |
| 13 | Gathering Compressors (C) | Emissions from compressor stations characterized as "gathering". Emissions constant among compressors. |
| 14 | Other Compressors (C) | Emissions from all other compressor stations. Emissions constant among compressors. |





**Table 2: Annual emission rate totals from anthropogenic sources within the innermost model domain based on values from the inventory within this study**

| Source | Total Emission Rate (Gg CH$_4$ year$^{-1}$) |
|---|---|
| Unconventional Wells | 125 |
| Conventional Wells | 607 |
| Gathering Compressor Facilities | 118 |
| Storage Facilities | 69 |
| Gathering/Transmission Pipelines | 8 |
| Natural Gas Distribution | 213 |
| Underground, Surface, and Abandoned Coal Mines | 831 |
| Enteric Fermentation/Manure Management | 371 |
| Landfills | 420 |
| **Total** | **2762** |

**Table 3: Meteorological statistics from the May 2015 flight campaign.**

| Day | Flight Pattern | # of Loops | # of Vertical Profiles | ABL Depth (m) | Mean Observed Wind Speed (m/s) | Mean Observed Wind Direction | Model Background Value (ppm) |
|---|---|---|---|---|---|---|---|
| May 14 | Box | 1 | 2 | 1300 | 2.9 | 30° | 1.908 |
| May 21 | Raster | N/A | 2 | 1300 | 3.9 | 231° | 1.905 |
| May 22 | Box | 2 | 2 | 2300 | 10.1 | 300° | 1.910 |
| May 23 | Box | 2 | 2 | 1400 | 4.4 | 276° | 1.906 |
| May 24[1] | Other | N/A | 2 | 1500 | 4.4 | 270° | 1.923 |
| May 24[2] | Raster | N/A | 2 | 2050 | 4.8 | 272° | 1.907 |
| May 25 | Box | 1 | 2 | 1800 | 9.0 | 217° | 1.920 |
| May 28 | Box | 2 | 3 | 1400 | 7.1 | 322° | 1.897 |
| May 29 | Box | 2 | 2 | 1000 | 4.6 | 195° | 1.899 |
| June 3 | Raster | N/A | 1 | 1250 | 2.7 | 149° | 1.898 |





**Table 4: Optimized natural gas emission rates for each flight as well as corrected emission rates adjusting for errors in the model wind speed and boundary layer height  For wind speed and boundary layer height error, a negative value represents a model value less than the observations.**

| Day | Optimized NG Emission Rate (% of production) | Wind Speed Error (6) | Boundary Layer Height Error (7) | Corrected NG Emission Rate (% of production) |
|---|---|---|---|---|
| May 14 | 0.37 | -31% | -33% | 0.17 |
| May 21 | 0.53 | 3% | 39% | 0.76 |
| May 22 | 1.15 | 37% | -18% | 1.30 |
| May 23 | 0.45 | 34% | -9% | 0.55 |
| May 24 | 0.68 | 48% | -21% | 0.80 |
| May 24 | 0.36 | 48% | -21% | 0.42 |
| May 25 | 0.99 | 3% | -43% | 0.58 |
| May 28 | 0.33 | -4% | -8% | 0.29 |
| May 29 | 0.35 | 4% | 1% | 0.37 |
| June 3 | 0.26 | 19% | -8% | 0.29 |
| **Average** | **0.55** | **16%** | **-12%** | **0.55** |

**Table 5: Emission rates and potential errors associated with the model optimization technique**

| Day | Optimized Upstream Emission Rate (% of production) | Background Error | Non-Upstream Gas Inventory Error | Model Performance Error | Total Error | 2σ Confidence Interval (% of Production) |
|---|---|---|---|---|---|---|
| May 14 | 0.37 | ±24% | ±20% | ±17% | ±36% | ±0.13 |
| May 21 | 0.53 | ±24% | ±17% | ±30% | ±42% | ±0.22 |
| May 22 | 1.15 | ±38% | ±6% | ±37% | ±53% | ±0.61 |
| May 23 | 0.45 | ±39% | ±13% | ±42% | ±59% | ±0.26 |
| May 24[1] | 0.68 | ±24% | ±54% | ±17% | ±61% | ±0.42 |
| May 24[2] | 0.36 | ±51% | ±78% | ±31% | ±98% | ±0.35 |
| May 25 | 0.99 | ±29% | ±19% | ±30% | ±46% | ±0.45 |
| May 28 | 0.33 | ±76% | ±36% | ±20% | ±86% | ±0.29 |
| May 29 | 0.35 | ±24% | ±9% | ±19% | ±32% | ±0.11 |
| June 3 | 0.26 | ±31% | ±10% | ±24% | ±40% | ±0.11 |





**Table 6: Emission rates from mass balance calculations on applicable days, with emission ranges associated with a ±5ppb error in the background value.**

| Flight | $CH_4$ Production within box ($Gg\ hr^{-1}$) | Mass Balance $CH_4$ Flux ($kg\ hr^{-1}$) | Non-Upstream $CH_4$ Emissions ($kg\ hr^{-1}$) | Calculated Upstream Emission Rate (% of production) | 2σ Confidence Interval (% of Production) |
|---|---|---|---|---|---|
| May 22$_1$ | 4.96 | 53800 | 2250 | 1.04 | ±1.09 |
| May 22$_2$ | 4.96 | 27400 | 2250 | 0.51 | ±1.08 |
| May 23$_1$ | 4.05 | 5600 | 934 | 0.11 | ±0.07 |
| May 23$_2$ | 4.05 | 5500 | 934 | 0.11 | ±0.07 |
| May 28$_1$ | 3.73 | 7100 | 706 | 0.17 | ±0.11 |
| May 28$_2$ | 3.73 | 6000 | 843 | 0.14 | ±0.10 |
| May 29$_1$ | 4.63 | 27900 | 1622 | 0.57 | ±0.30 |





**Table 7: Relative error associated with the different sources of uncertainty.**

| Flight | Wind Speed Error | Background Error | ABL Error | Inventory Error | Total Error (1σ) | Upstream Emission Rate (% of Production) w/ 2σ Confidence Interval |
|---|---|---|---|---|---|---|
| May 22$_1$ | ±3% | ±56% | ±9% | ±5% | ±57% | 1.04 ±1.09 |
| May 22$_2$ | ±3% | ±121% | ±9% | ±8% | ±121% | 0.51 ±1.08 |
| May 23$_1$ | ±3% | ±24% | ±7% | ±20% | ±32% | 0.11 ±0.07 |
| May 23$_2$ | ±3% | ±26% | ±7% | ±21% | ±34% | 0.11 ±0.07 |
| May 28$_1$ | ±3% | ±31% | ±7% | ±11% | ±34% | 0.17 ±0.11 |
| May 28$_2$ | ±3% | ±33% | ±7% | ±16% | ±38% | 0.14 ±0.10 |
| May 29$_1$ | ±3% | ±28% | ±20% | ±8% | ±36% | 0.57 ±0.30 |

**Table 8: Production statistics from mid-2014 for various shales across the United States (Hughes 2014).**

| | Barnett | Fayetteville | Haynesville | Marcellus | Bradford/ Susquehanna County, PA |
|---|---|---|---|---|---|
| # of Producing Wells | 16100 | 4500 | 3100 | 7000 | 1558 |
| Total Production (Bcf day$^{-1}$) | 5.0 | 2.8 | 4.5 | 12 | 5.01 |
| Production per well (MMcf day$^{-1}$) | 0.31 | 0.56 | 1.25 | 1.71 | 3.22 |





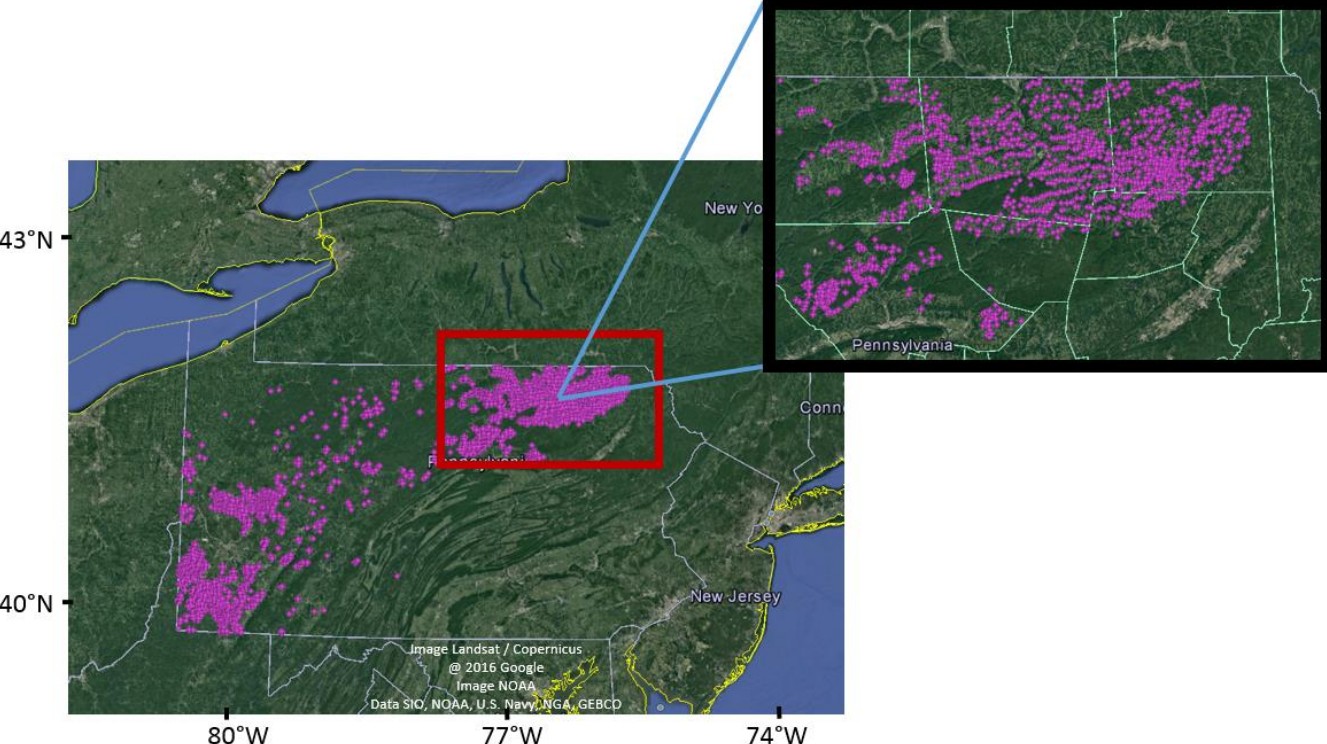

**Figure 1: A map of the unconventional wells in Pennsylvania dotted in purple. Red rectangle and zoom-in show the region of focus for this study, 41.1-42.2°N 75.2-77.6°W.**



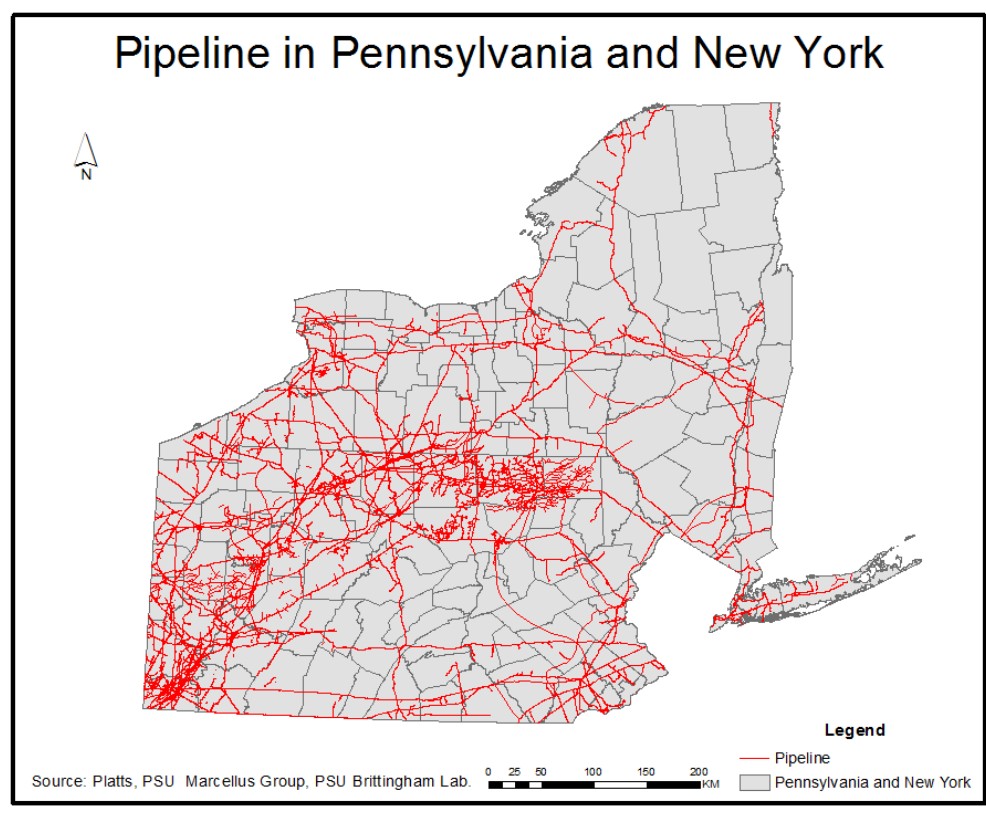

**Figure 2: A map of transmission and gathering pipelines for the state of PA and NY. Transmission pipelines are provided by Platts Natural Gas Pipelines product. Gathering pipelines associated with unconventional wells in PA are extrapolated using information on existing gathering pipelines provided by Bradford County, PA.**




**Figure 3: Observed CH₄ enhancements within the boundary layer from each of the 10 afternoon flights used in this study, with green dots showing the location of unconventional wells in PA and blue arrows showing the modelled wind direction during the time of the flight. CH₄ enhancements are calculated by taking the observed CH₄ mole fraction values and subtracting off the flight's background CH₄ value shown in Table 3.**





**Figure 4: Model domain and resolutions used within the transport model. All emissions used for this study are contained within the 3 km resolution domain.**



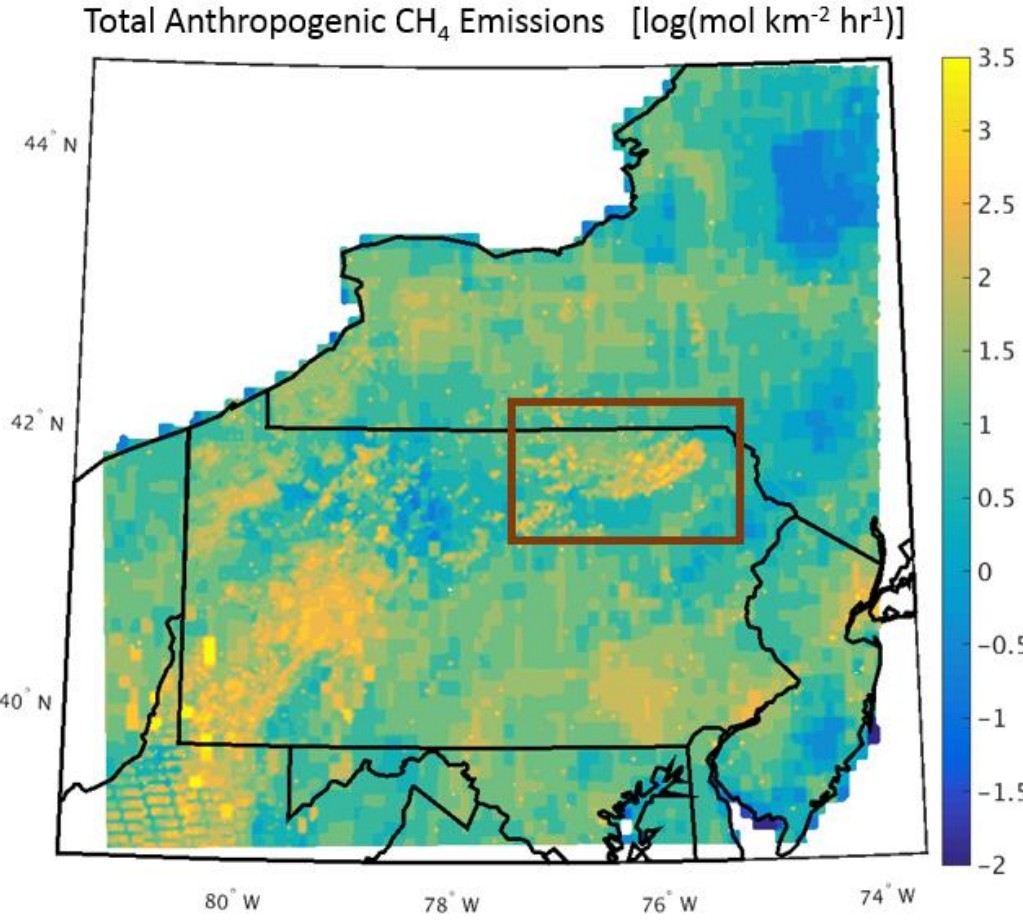

**Figure 5: A log scale contour of the anthropogenic CH₄ emissions inventory from this study used within the transport model. The red rectangle surrounds the study region where the aircraft campaign took place.**



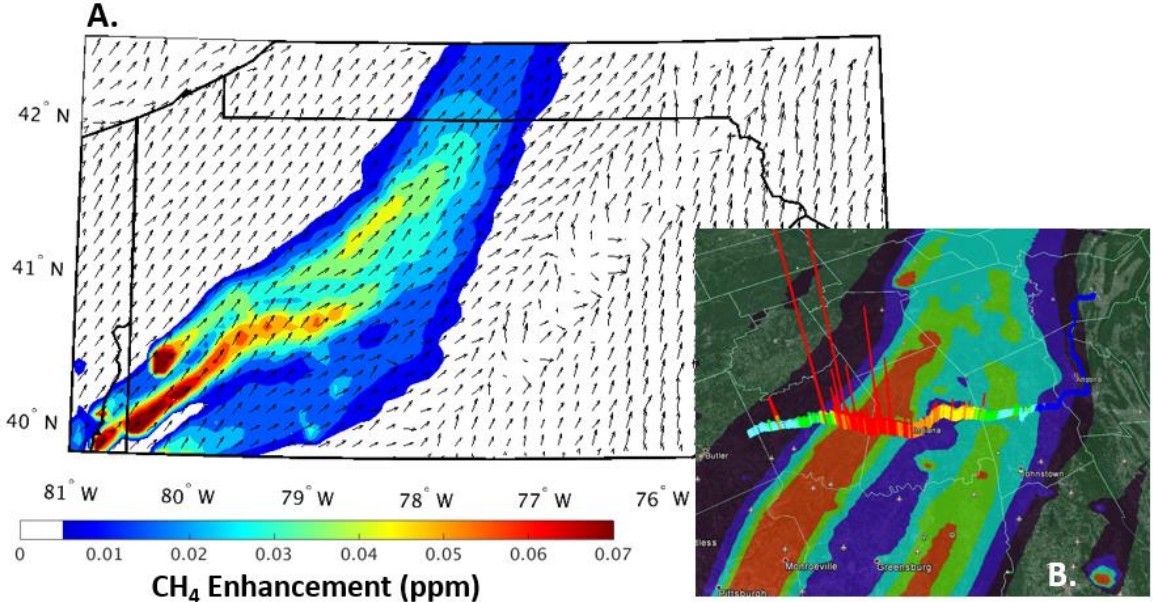

**Figure 6: (a.) Model projected CH$_4$ enhancement at the surface associated with underground, surface. and abandoned coal mines on May 27$^{th}$, 2015 at 19Z, with the shaded regions showing the CH$_4$ enhancement and the arrows representing the wind direction. (b.) Projected enhancement from a. mapped over measured CH$_4$ enhancement from a driving campaign. The height and colour of the bars represents the scale of the CH$_4$ enhancement.**



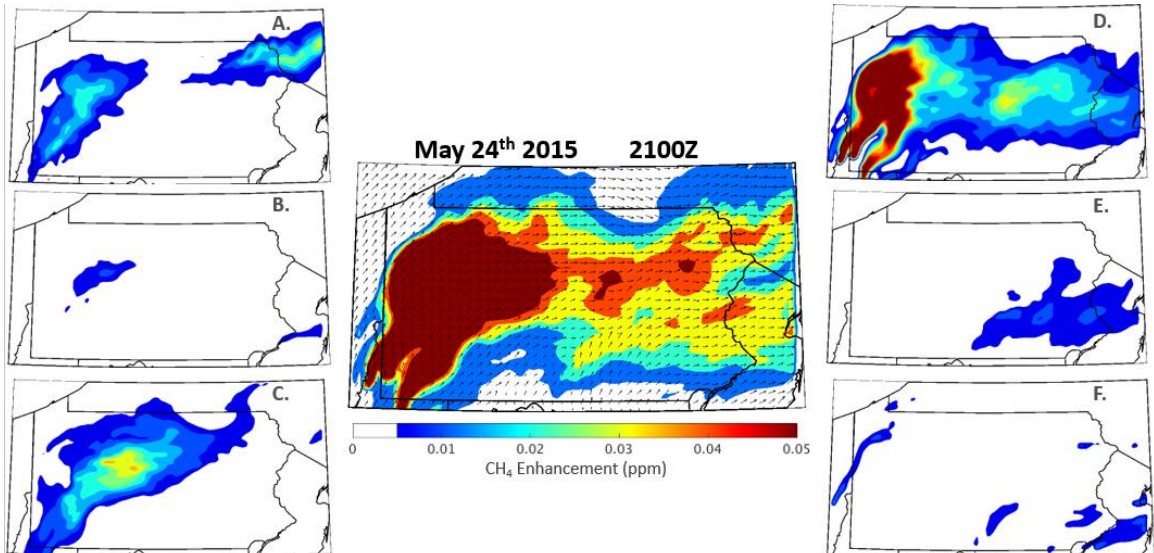

**Figure 7: Projected CH₄ enhancements during the late afternoon flight of May 24th, 2015 at 2100Z, 700m above ground level from (A) upstream unconventional gas processes (B) downstream unconventional gas processes (C) conventional production (D) coal mines (E) animal emissions and (F) landfills and other sources within the EPA GHG Inventory Report. The centre figure is a map of the combined enhancement from sources A-F.**



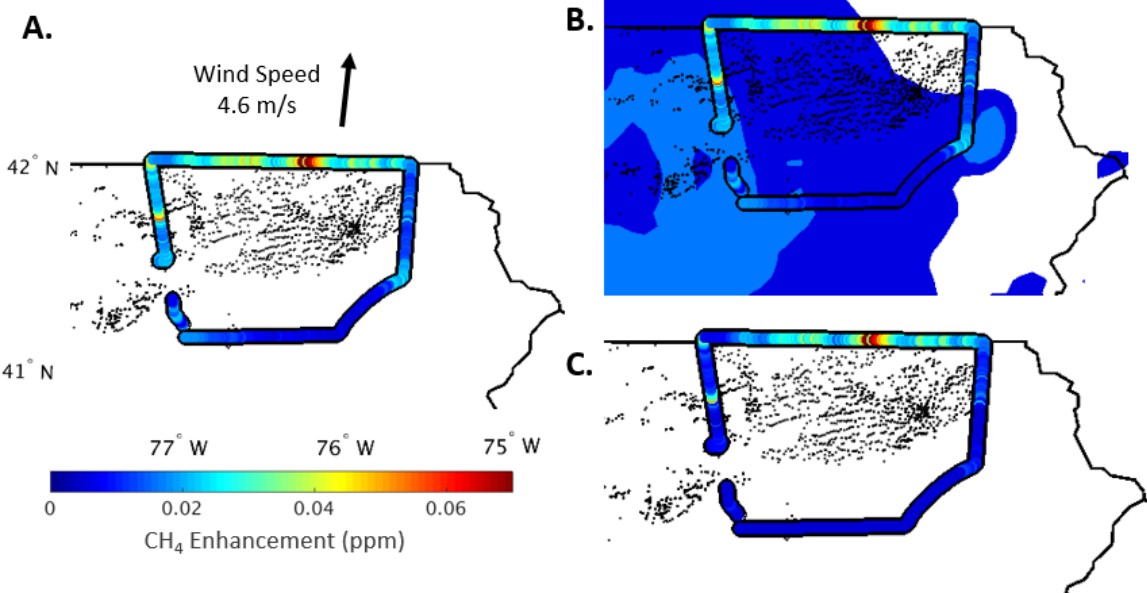

**Figure 8:** (a.) Observed CH₄ enhancements from within the boundary layer during the first loop of the May 29th aircraft campaign. (b.) Aircraft observations laid overtop modelled CH₄ concentrations from sources unrelated to emissions from upstream gas production. (c.) Observed CH₄ enhancements from the May 29th flight after subtracting off modelled sources in b. The new set of observations represent the observed upstream gas enhancement during the flight.



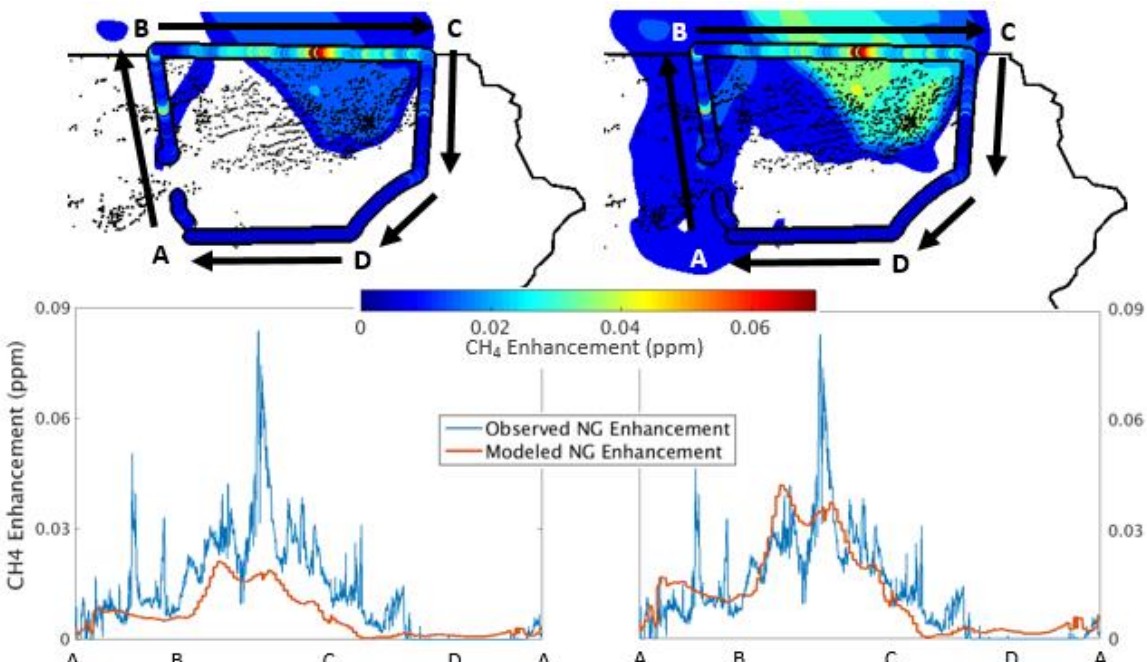

**Figure 9: (top-left) Observed enhancement from unconventional natural gas production overtop projected upstream natural gas enhancements from the first loop of the May 29th flight, using an upstream gas emission rate of 0.13% of production. (bottom-left) Direct comparison of the observed natural gas enhancement vs. the modelled enhancement following the path from A-D using an unconventional emission rate of 0.13%. (top-right, bottom-right). Same as left figures, except using the optimized upstream emission rate of 0.26%**

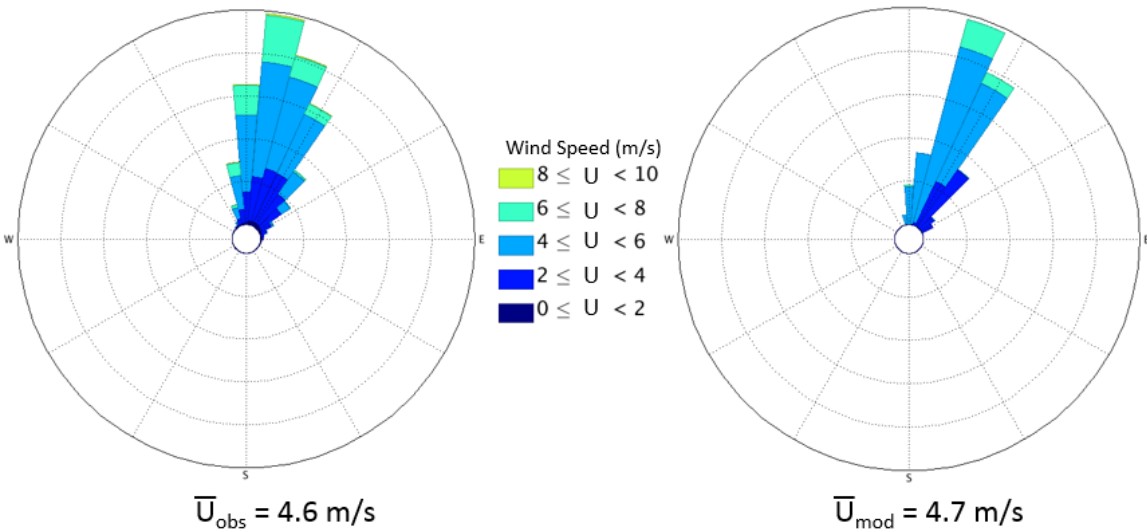

**Figure 10: Wind rose of aircraft observations (left) within the boundary from the first loop of the May 29th flight compared to modelled winds following the flight path (right).**



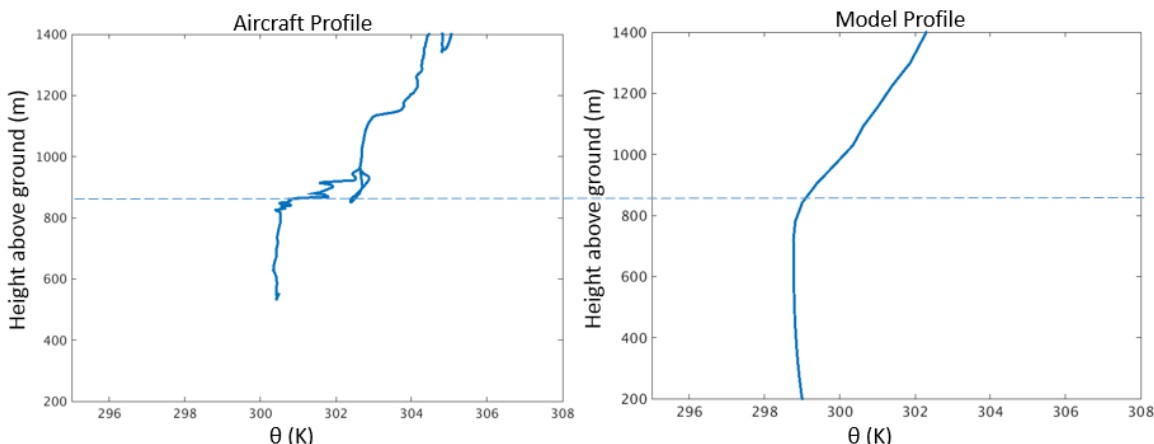

**Figure 11: (left) Observed potential temperature profile with height from the first aircraft spiral on the May 29th flight at 17Z. (right) Modelled potential temperature at the location and time at which the aircraft spiral occurred. In both cases, an inversion in the potential temperature profile begins to occur around 850m.**

## May 24th 2015: Late-Afternoon Flight

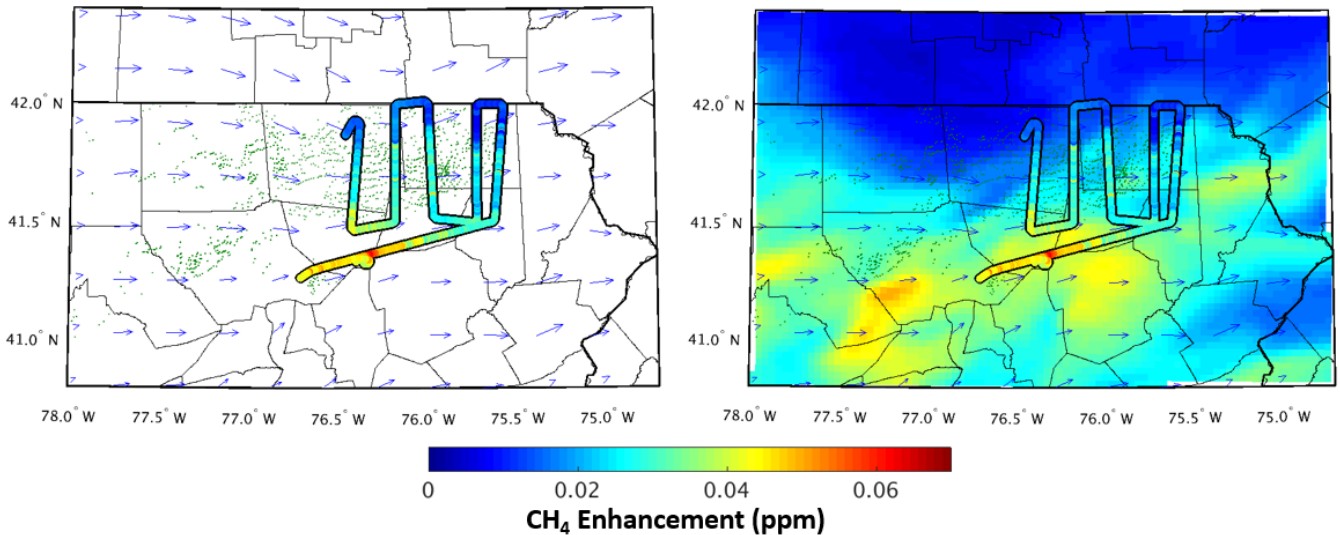

**Figure 12: (left) Observed CH₄ enhancement from the late-afternoon flight on May 24th, 2015. (right) Observed CH₄ enhancement compared to the model projected CH₄ enhancement from the sum of all sources in the region. The colour scale of observed and projected enhancements is scaled 1:1, with matching colours indicating matching values.**



**Figure 13: Observed CH4 enhancements from an early flight on May 28th, 2015 compared to projected CH4 enhancements from coal emissions modelled at (top) 14:00Z and (bottom) 15:00Z. The one hour time difference results in vastly different projected enhancements across the southern portion of observations.**





**Figure 14: Observed vs model projected CH₄ enhancements during (top) the early afternoon flight of May 24ᵗʰ, 2015 at 17Z and (bottom) the flight of May 25ᵗʰ, 2016 at 19Z.**





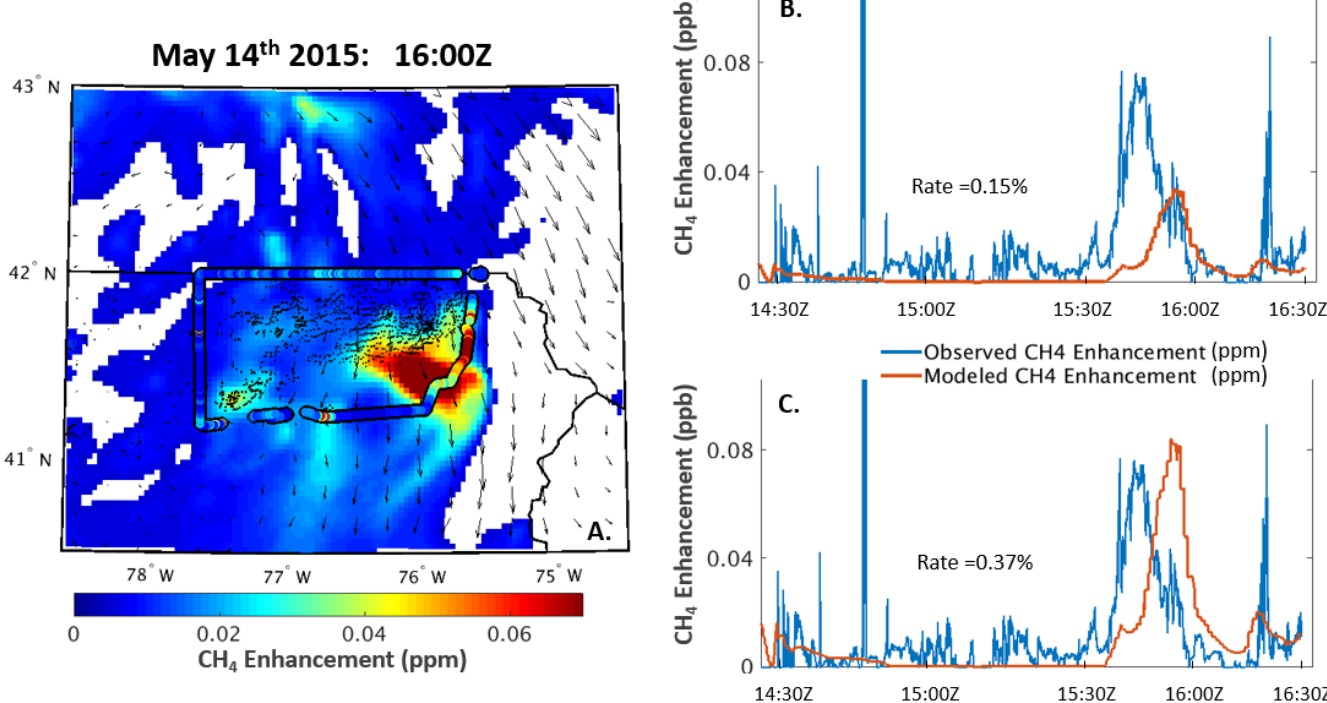

**Figure 15: (a.) Observed vs model projected CH₄ enhancements during the May 14th, 2015 at 16Z. (b.) Comparison of observed natural gas enhancement to modelled natural gas enhancement along flight path, with upstream emission rate optimized by minimizing the absolute error between the datasets. (c.) Same as previous, but optimized by minimizing the sum of the error between the datasets.**




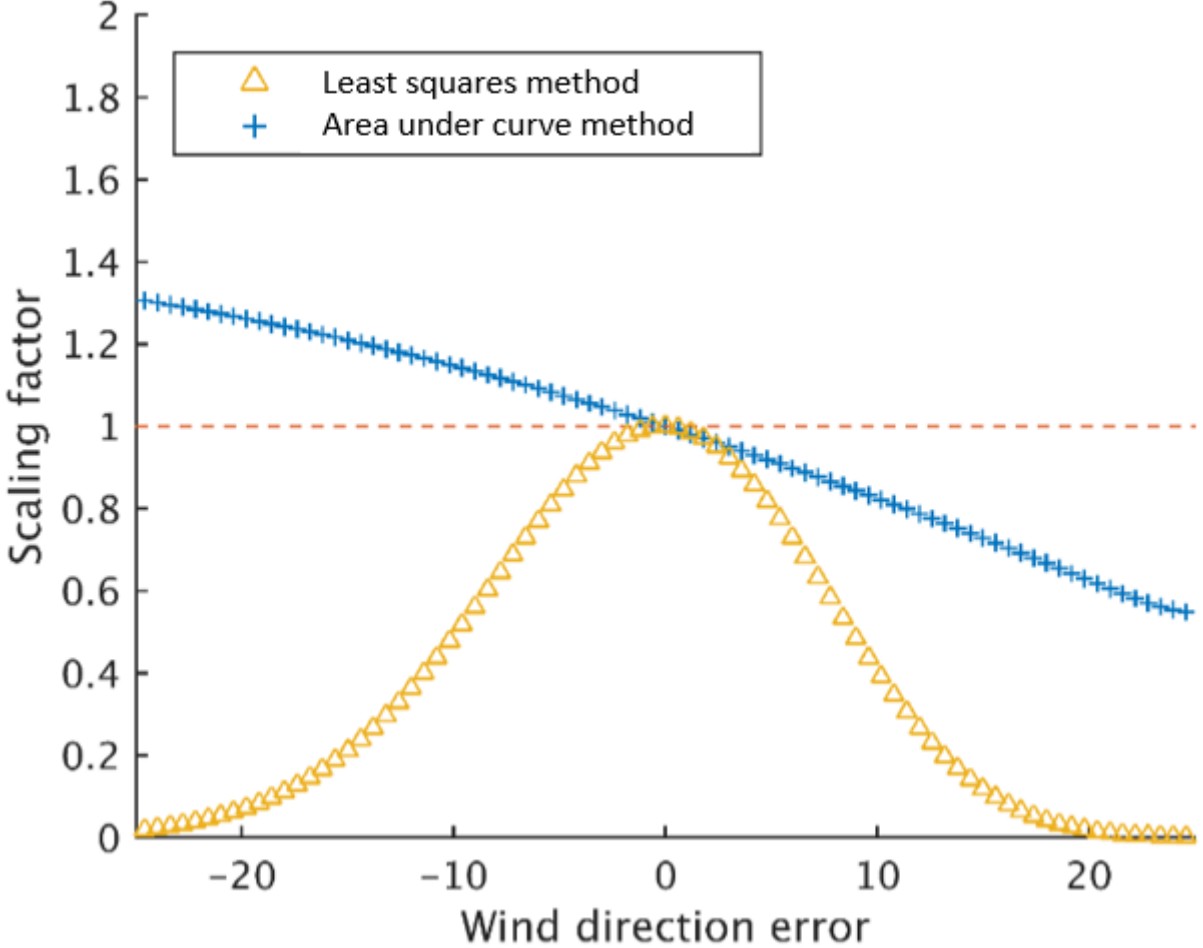

**Figure 16: A demonstration of the effect of model transport error on the optimized emission rate using pseudo-observations transecting a modelled plume at a 45° angle. The scaling factor represents the change in the optimized emission rate compared to the true emission rate, with 1 representing a perfect rate.**





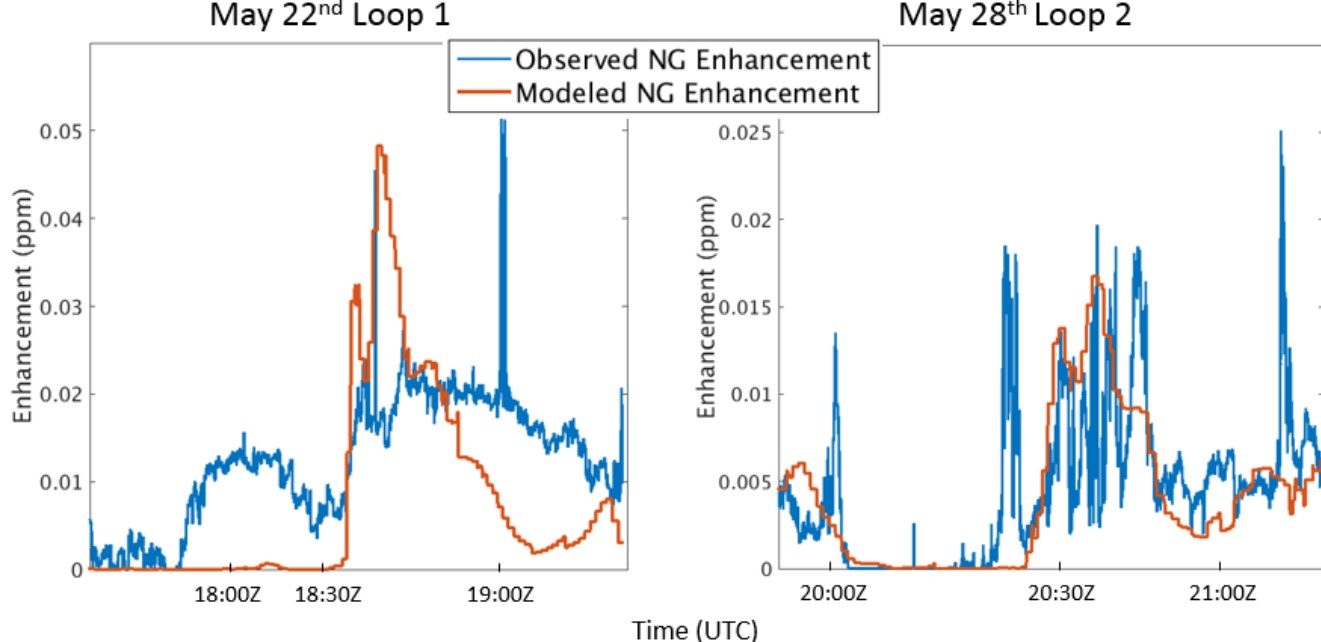

**Figure 17: Comparison of observed natural gas enhancement to modelled natural gas enhancement for segments along the (left) May 22nd flight and (right) May 28th flight. A distinct lack of representativeness of the observations in the modelled enhancement can be seen in the May 22nd flight compared to the May 28th flight.**




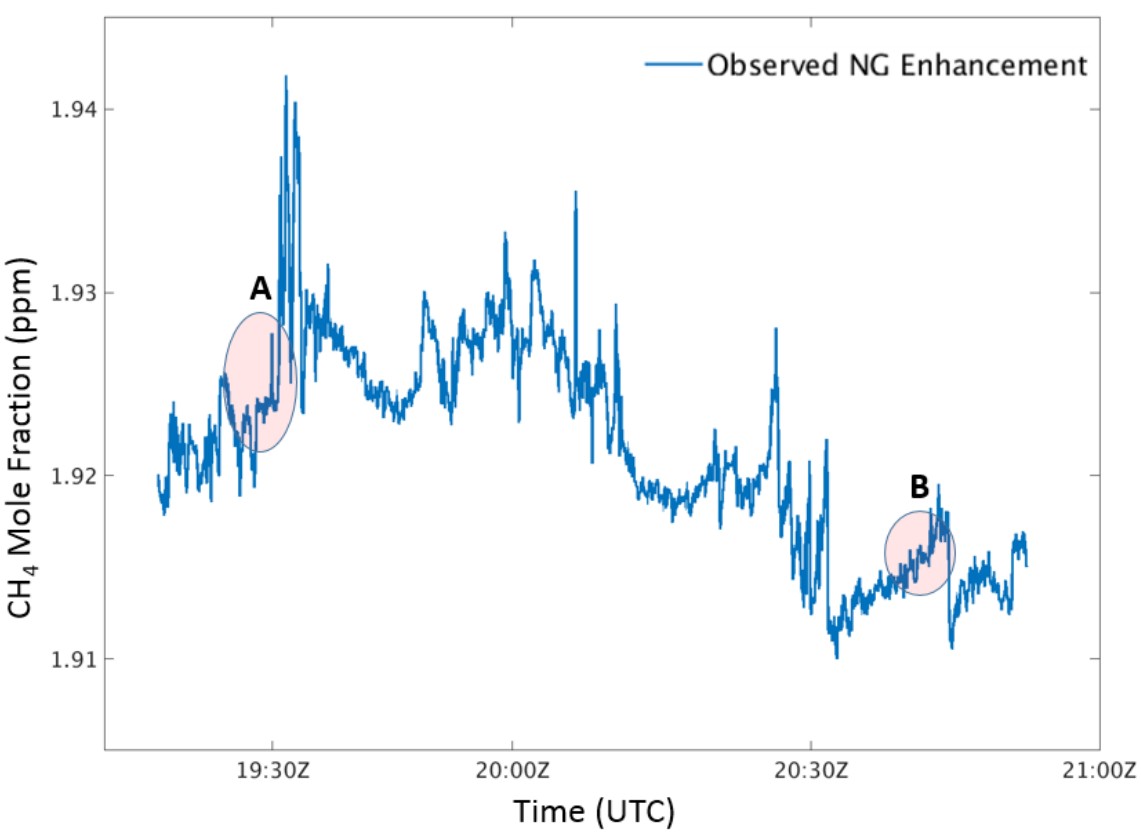

**Figure 18: Time series of CH₄ mole fractions from the second loop of the May 22ⁿᵈ flight. Observations at the shaded areas below A and B were taken at similar locations in space, showing the change in the background mole fraction across time.**





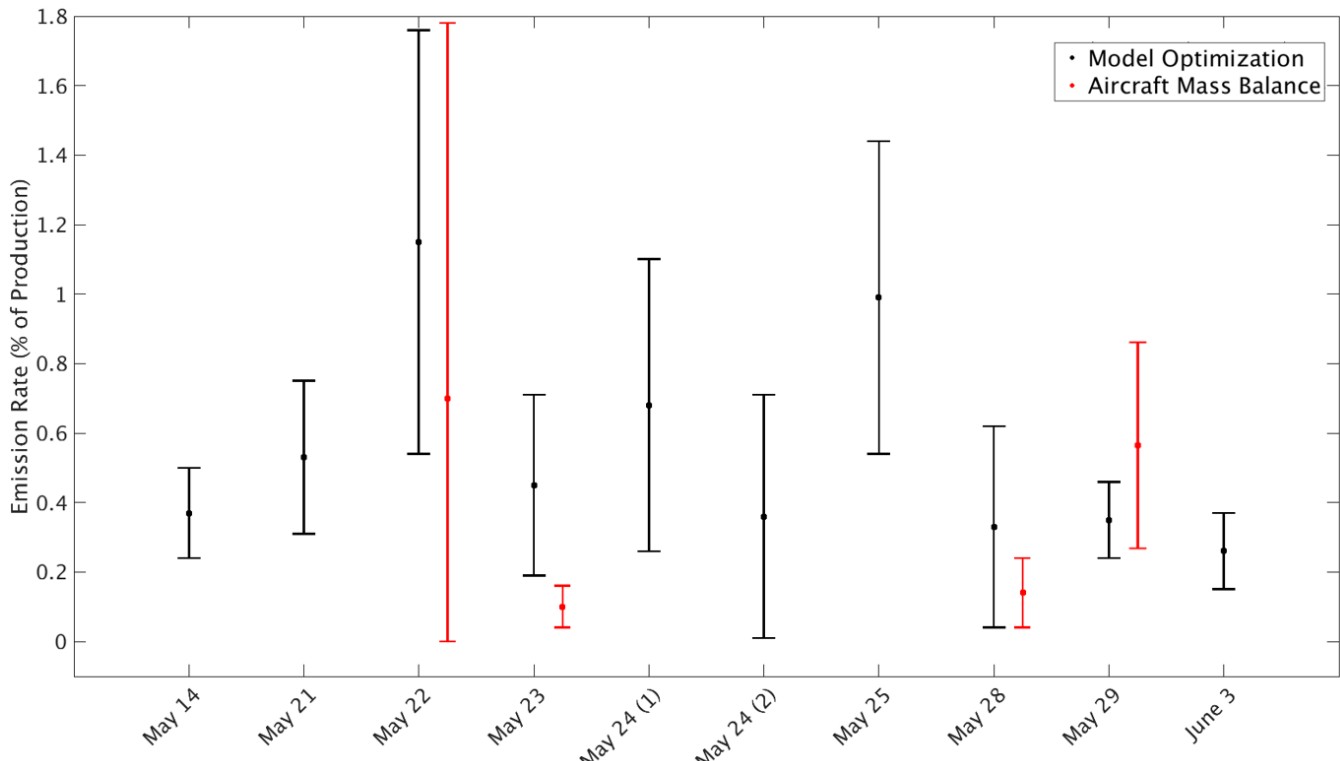

**Figure 19: Calculated upstream natural gas emission rates using (black) model optimization technique and (red) aircraft mass balance technique.**



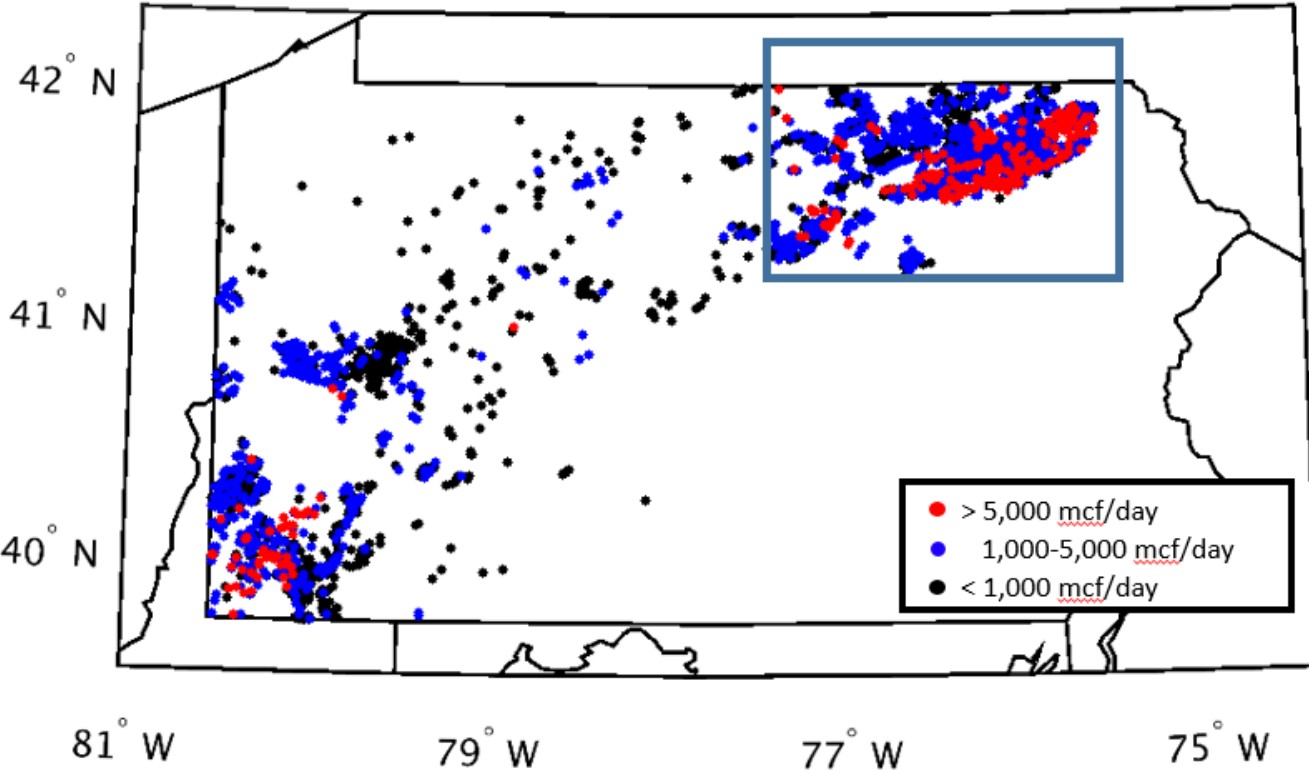

**Figure 20: Well locations and daily production of unconventional wells in PA for May 2015. Boxed region is study region where flights took place.**





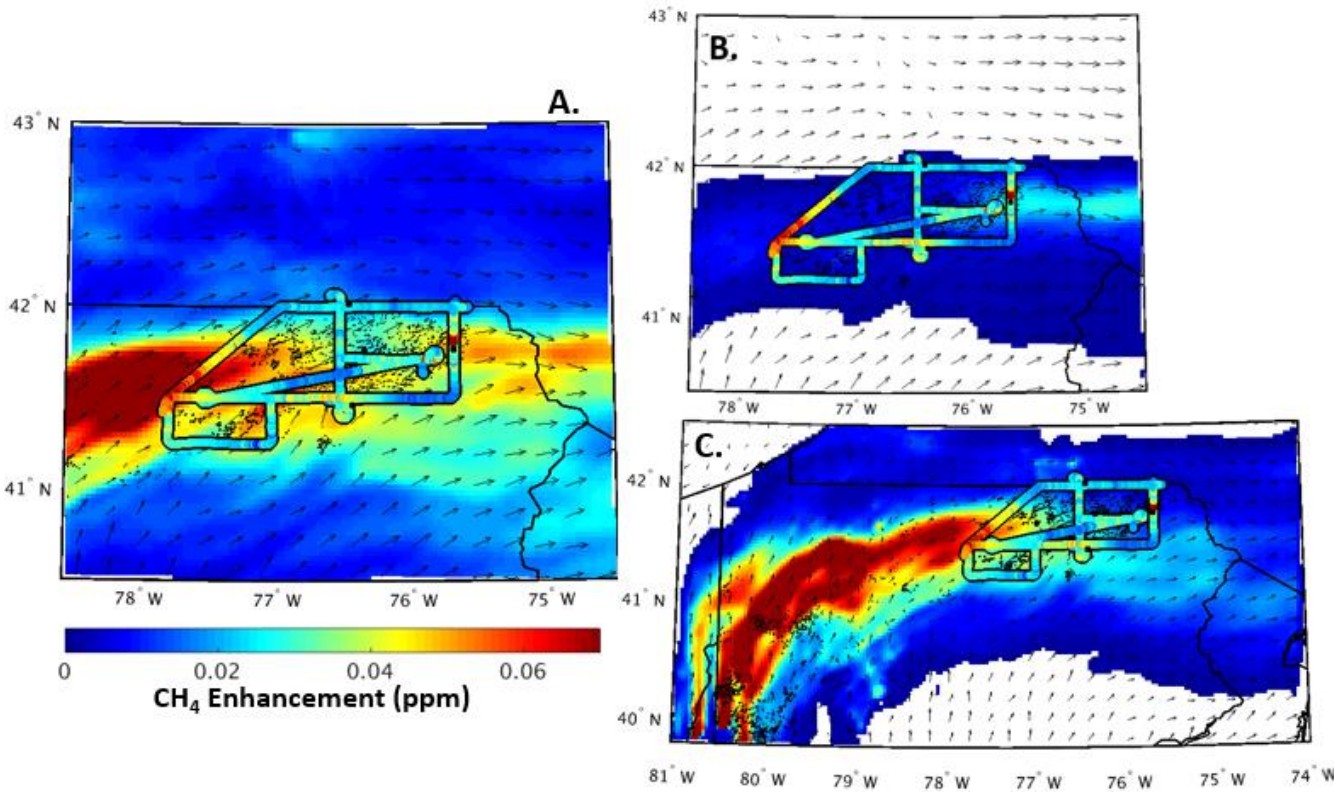

**Figure 21: Observations vs modelled enhancements of the flight from Peischl et. al (2015) for July 6th, 2013. (a.) Observed enhancements from the flight over model projected enhancements from all sources at 21Z. (b.) Projected enhancement from upstream gas processes using a 0.4% emission rate. (c.) Projected enhancement from coal sources in southwestern PA.**