# Peer review of "Quantifying methane emissions from natural gas production in northeastern Pennsylvania"

_Atmospheric Chemistry and Physics, 2017_

## Referee Comment (RC1) · Anonymous Referee #1 · 25 May 2017

Barkley et al. use measurements of methane from an aircraft to quantify emissions from the Marcellus shale area of northeast Pennsylvania. Their analysis includes both a modeled emission estimate using 10 flights in May 2015, as well as an "aircraft mass balance" estimate using 4 of those same flights. When presented as a percentage of production, the modeled emissions are 0.36% (2-sigma 0.27-0.45%) and the aircraft mass balance emissions are 0.34% (2-sigma 0.06-0.62%).

Main comments

This is a well-written paper, and the model seems to reproduce the various sources of CH4 in Pennsylvania fairly well. There is also good agreement between the modeled emission and the mass balance determined emission.

[Figure]

Other comments

line 109-112: what are the range of estimates for these numbers? You provided them for the Karion studies, but not these other two.

line 184: Perhaps cite the latest EPA greenhouse gas inventory.

line 260-261: do these CH4 sinks account for other reactive species in the atmosphere? If not, how would background or elevated levels of ethane or other gases affect these CH4 losses?

Figure 2: if possible, it would be helpful to know where Bradford County was on this map.

Figure 19: I would add the mean model optimization and aircraft mass balance emission rates to this graph, as well as the Omara and Peischl studies, along with their uncertainties. It seems like one advantage this study has, as mentioned in the introduction, is a narrowing of the uncertainties of these previous studies.

Technical/grammatical comments

The author numbering is not in order.

line 48: Methane also has an atmospheric sink from chlorine chemistry.

line 58: is "mining" the proper word for natural gas? It may be, I just can't say I've heard that before.

line 64: add "of" to say "quality and quantity of its emission factors".

line 107: put "per day" before the abbreviation "MMSCFD", instead of after it.

———————————————

---

## Referee Comment (RC2) · Anonymous Referee #2 · 8 Jul 2017

The manuscript "Quantifying methane emissions from natural gas production in north-eastern Pennsylvania" by Barkley and co-authors presents CH4 emission rate estimates for a natural gas extraction area in the north-eastern United States. The authors use observations from 10 individual flights, high-resolution CH4 transport simulations in combination with an optimisation approach (inverse modelling) and mass balance calculations for their estimates. Although, there is large interest in such independent emission rate estimates throughout the US, the presented results somewhat fail to convince that the applied methods are suitable for a robust estimation. Especially the presented uncertainty assessment remains largely arbitrary and needs further improvement. In addition , the manuscript should be shortened and needs a number of

small and technical corrections.

Major comments:

I have three major areas of concern that are briefly mentioned here, but more details follow below along the course of the manuscript.

Model optimisation technique (inverse modelling): The applied optimisation technique uses a very simple cost function to find an optimal factor between 'prior' and 'posterior' emission rates. Why did you not apply a Bayesian approach that could take uncertainties in the prior values and observations/simulations into account? Much of the following uncertainty assessment could then be used in the optimisation step and also more reliably give posterior uncertainties. Also, why do you not use sum of squares in the cost function as is commonly done? Some explanation is given later in the text, but without further discussion of Figure 16 this remains useless. Please discuss why your method should provide more reliable results as other similar studies that have used a Bayesian framework.

Uncertainty assessment of obtained emission rates: The uncertainty assessment of the obtained emission rates for both the 'model optimisation' as well as the mass balance method contains a lot of arbitrary assumptions and does not seem to be statistically sound. Their seems to be an overall assumption of Gausssian uncertainties (although not explicitly mentioned), which does not seem to be justified for several of the sources of uncertainty. Also the way the uncertainties are propagated through the individual methods remains very vague and could be improved by using a Monte Carlo type uncertainty assessment. These concerns are repeated in detail below.

Length: With 21 figures, 8 tables and a total of 50 draft pages the manuscript is quite lengthy and it would profit from shortening and restructuring. Below I suggest a number of figures that could easily be omitted or moved into a supplement without loss of information. Furthermore, I strongly encourage to incorporate the description of the uncertainty assessment method, which is now given within the results section, into

[Figure]

the methods section along with or following the description of emission rate estimation itself. In this was many repetitions could be avoided and the paper could really focus onto the results in section 3.

Minor comments:

L51: Shouldn't you rather say that for emission rates larger than 3 %? Otherwise the sentence does not define a threshold.

L232-238: This paragraph should be directly in front of the one describing the data assimilation strategy finally used in the study (L246-253).

L261: The important time scale for the mass balance approach is not alone the flight duration but also the transport time from the upwind to the downwind interface of the box. Also the time from emission to sample should be discussed, as this is the crucial one for the inverse modelling.

L312 and elsewhere: I don't like the terminology here. "observed CH4 enhancement associated with upstream natural gas". The described quantity contains information from both model and observation and, thus, is not purely 'observed'. Maybe the term 'observation-based' or something else that identifies the real character of the data could be used instead.

L321: How frequent and how large are negative values of X_GasO?

L352: Were individual atmospheric densities for each flight neglected? Why?

L416: If this is a good correlation coefficient, what is the range obtained for the other flights? Could this information be added to Table 4 or 5. It could also be used as part of the uncertainty assessment. See comment below.

L418f: Be a bit more specific concerning the near zero differences between model and observation. For the PBL height this statement may be true, however, it is also clear that the model is not able to capture the very sharp increase in potential temperature. I

also wonder why the two profiles are shown separately, would be nice to have them on top of each other. Concerning the wind there is a directional shift of about +10 degree in the model. You may call this small, but for the kind of plumes simulated here it might be of importance and should be mentioned.

L424ff: Please add a number for the average and maximum enhancement due to upstream natural gas sources at the northern transect to this discussion. From Figure 9 I would think these numbers are around 30 and 80 ppb. It puts the other contributions into perspective.

L473ff: This discussion on the uncertainties due to shifts in the simulated wind direction is very hard to follow. Figure 16 does not speak for itself. You will need to explain in more detail what was done to derive the figure. This discussion should actually given when the cost function is introduced.

L486: Should it be wind speed instead of wind direction?

L498ff: The way the wind speed error is assessed biases could be introduced. The average over all wind observations by the aircraft does not necessarily reflect the average wind speed within the boundary layer, since the sampling is not uniform (especially with height). It seems calculating an average vertical profile of wind speed from all the observations and comparing that to an average model profile generated from the same sampling locations should be more robust. Was this considered and decided that the aircraft sampling was sufficiently uniform?

L506ff: There are more advanced techniques for estimating boundary layer heights from model and observational data than just 'looking' at the potential temperature. For example Bulk Richardson methods are quite useful in situations when potential temperature alone is not providing explicit results. The example shown here is an easy case but I wonder about the other situations mentioned in the text and especially how large the uncertainty of the estimates becomes in such situations.

L542f: How was this done exactly? Did you simply run 3 cases with 1) reference background, 2) background -5 ppb, 3) background +5 ppb? Or did you do a Monte Carlo approach where the +/- 5 ppb could be understood as the standard deviation of a normal distribution? Please clarify and justify the approach also in the light of possible non-linear effects that would not be covered by simply running 3 cases.

L548f: It is mentioned that the +/- 5 ppb assumption for the background uncertainty can have different impacts depending on the magnitude of the observed plume and this is related to wind speeds and PBL heights. How can you justify the use of a constant value for the background uncertainty? Shouldn't this also change with meteorological conditions? For example it is mentioned earlier that it will be large when there is more entrainment, which is usually the case while the PBL is still developing, and when plumes from other sources are adveced. Wouldn't it make sense to take this variability in the background uncertainty into account?

L557f: Again the question: is this done by running by simply running 3 cases or in a Monte Carlo fashion. Did you scale all categories with the same factor at the same time or test combinations as well? How do you obtain symmetric uncertainties in the final emission rate when you use asymmetric uncertainties in the non-upstream natural gas emissions?

L505, L562: Table 5 should already be referenced here and not only at the very end of the section.

L571: The estimator for the model performance uncertainty is a normalised root mean square error (NRMS) and it should be called like this to reduce the confusion. However, I am not convinced that this is the best estimator for what you want to achieve. One problem is the use of the optimised simulation, which means you have already used the information contained in the observations and thus the uncertainty estimator is not independent anymore. Furthermore, NRMS is not a good measure of what you call "similarity of pattern". The Pearson correlation coefficient should be better suited for

this purpose. Please justify your choice. If you stay with a RMS estimation it should be applied to the prior simulations and the bias should be removed before the RMS is estimated.

L579f: You also assume Gaussian uncertainty distributions, which seems oversimplified.

L645: Figure 18 can go into a supplement without any loss of information. Instead give the rate of change of the background with time as a number in the text.

L652: Does this mean the final uncertainty estimate is the mean over the uncertainties for each flight?

L663ff and Fig. 19: The variability in emission rate estimates for individual days as shown in the figure are not discussed in any detail. Of special interest is the question of agreement between the two methods as it could lend some additional trust in the applied methods. However, the figure seems to raise more questions in the reliability. For example if 29 May is the golden day (as described earlier in the text), why are estimates from both methods so different? Also there seems to be no agreement in the variability for the 4 common days. How could you explain the large variability from the point of the emission processes? Is there any dependency on the total production on these days (following the line of thought used later for the discussion of different basins)? Please speculate on this otherwise the reader is left with the impression that your given uncertainty estimate is rather optimistic.

L666f: Repeat the range of emission rates from other studies here.

L696ff: The flight described in this section and shown in Fig. 21 is not part of the main analysis of the paper and should be omitted as it does not provide any additional insights and aspects that have not or could not be shown on the other flight examples.

Technical comments:

L26: Abbreviation (WRF-Chem) not introduced before. Do so on line 25.

L48: "transforms" instead of "transform".

L59: Add "United States" in front of "Environmental Protection Agency".

L348: Why are the terms U, D, and L given in braces? Not necessary.

L496: "as long as" instead of "so long as".

Fig. 7: Do not show wind arrows in center panel. Not visible anyway and one gets the general flow direction from the shape of the plumes. Show center panel with same size as the individual tracers. Also use exactly the same color scale for all sub-panels. It looks like the center plot does have fewer color levels than the others. Each panel (also the center panel) requires a label.

Fig. 11: Show observed and simulated profile on top of each other (see also comment above).

Fig. 12: There is no mentioning of the wind arrows in the figure legend. Are these near surface or mean PBL winds? There is also no reference vector that would allow inferring the wind speeds from the arrow length. Choose a different color for the wind vectors. They are not well visible in the righthand figure.

Fig. 13: There is no mentioning of the wind arrows in the figure legend. Are these near surface or mean PBL winds? There is also no reference vector that would allow inferring the wind speeds from the arrow length.

Fig. 14: There is no mentioning of the wind arrows in the figure legend. Are these near surface or mean PBL winds? There is also no reference vector that would allow inferring the wind speeds from the arrow length. Choose a different color for the wind vectors.

Fig. 15: There is no mentioning of the wind arrows in the figure legend. Are these near surface or mean PBL winds? There is also no reference vector that would allow inferring the wind speeds from the arrow length.

Fig. 19: Mention in the caption what error bars represent.

Fig. 20: This information could also be merged into Fig. 1 and save the additional figure here.

Fig. 21: There is no mentioning of the wind arrows in the figure legend. Are these near surface or mean PBL winds? There is also no reference vector that would allow inferring the wind speeds from the arrow length.

[Figure]

---

## Author Comment (AC1) · 18 Aug 2017

**Response to Reviewer 1:**

**Minor comments**

*line 109-112: what are the range of estimates for these numbers? You provided them for the Karion studies, but not these other two.*

Added the ranges of each to the paper.

*line 184: Perhaps cite the latest EPA greenhouse gas inventory*

We cited one from the previous year here because that was where the emission factor came from when this project was active. We cannot cite the most recent report here as the emission factors change from year to year and so the number may be different from the one we used.

*line 260-261: do these CH4 sinks account for other reactive species in the atmosphere? If not, how would background or elevated levels of ethane or other gases affect these CH4 losses?*

This is a fair point. Gases other than $CH_4$ are emitted from wells and could possibly decrease the reaction rate of OH with $CH_4$. However, the purpose of this section is just to show that even in above average OH concentrations, the $CH_4$ sink is negligible. Unless the other gases released from well emissions lead to the production of additional OH, any gas released that will react with OH will only lower the sink's impact further.

*Figure 2: if possible, it would be helpful to know where Bradford County was on this map*

Updated the figure with the county highlighted in yellow

[Figure]

*Figure 19: I would add the mean model optimization and aircraft mass balance emission rates to this graph, as well as the Omara and Peischl studies, along with their uncertainties. It seems like one advantage this study has, as mentioned in the introduction, is a narrowing of the uncertainties of these previous studies.*

I did not add the mean for each as it made the plot a bit heavy and I want to emphasize the spread for each day. I also did not add Omara's number as it comes from emissions only from well production (no gathering emissions) and all wells from that study were measured in the southwest Marcellus, which has a much different composition compared to the northeast Marcellus. However, I did add the number from the Peischl study to the plot, as his study is a single mass balance flight in the region and so it fits in perfectly with the rest of the graph's content. I am not sure our final findings narrowed the uncertainties as intended so much as it discovered how uncertain the previous estimate may have been.

[Figure]

**Technical Comments**

*The author numbering is not in order.*

Fixed

*line 48: Methane also has an atmospheric sink from chlorine chemistry.*

Clarified the sentence to imply the sink was mostly from OH.

*line 58: is "mining" the proper word for natural gas? It may be, I just can't say I've heard that before.*

Forget you heard it. It is not a good use of the word. Switched to "drilling and transportation of natural gas".

*line 64: add "of" to say "quality and quantity of its emission factors".*

Added.

*line 107: put "per day" before the abbreviation "MMSCFD", instead of after it.*

Switched the order.

---

## Author Comment (AC2) · 18 Aug 2017

**Response to Reviewer 2**

Minor comments are addressed first. Major comments which have not been addressed through fixes in the minor comments are then addressed afterwards.

**Many Minor Comments:**

*L51: Shouldn't you rather say that for emission rates larger than 3 %? Otherwise the sentence does not define a threshold.*

Changed to "greater than"

*L232-238: This paragraph should be directly in front of the one describing the data assimilation strategy finally used in the study (L246-253).*

Moved the paragraph accordingly

L261: *"The important time scale for the mass balance approach is not alone the flight duration but also the transport time from the upwind to the downwind interface of the box. Also the time from emission to sample should be discussed, as this is the crucial one for the inverse modelling."*

This segment is discussing how oxidation can decrease the background concentration of $CH_4$ during the duration of the flight. The emissions should not matter in this calculation because OH would be reacting with the entire $CH_4$ mole fraction (i.e., the background + enhancement), not just any enhancement in the region (a 40 ppb enhancement would change the CH4 mole fraction by 2%, so changes to the reaction rate would be minimal). So it will be affecting all parts of the flight almost equally throughout the time of the flight and shouldn't have any relation to the distance the emissions are travelling from the well to the downwind transect.

However, I do agree that it is good to know the time it takes for emissions to travel to the downwind transect, if only to better understand how representative the upwind transect is of the air mass measured by the downwind transect. Given the range in wind speeds, it takes about 2-4 hours. This is mentioned briefly in the error analysis section.

*L312 and elsewhere: I don't like the terminology here. "observed CH4 enhancement associated with upstream natural gas". The described quantity contains information from both model and observation and, thus, is not purely 'observed'. Maybe the term 'observation-based' or something else that identifies the real character of the data could be used instead.*

The term "observation-derived" replaces now "observed" across the entire document.

*L321: How frequent and how large are negative values of X_GasO?*

Added sentence in paper describing negative values. 16% of the observation-derived enhancements are negative, but less than 3% are negative by 5 or more ppb.

*L352: "Were individual atmospheric densities for each flight neglected? Why?"*

They were not calculated, because a +-10 change in the temperature produces a difference in the value (and thus the end result) of 3%. It's a similar change with a +-30 hPa change in the pressure. For 10 flights during a 3 week period with no huge meteorological anomalies, calculating the exact density of the air in the ABL (which will have its own errors) based on a few vertical transects seemed unnecessary.

*L416: If this is a good correlation coefficient, what is the range obtained for the other flights? Could this information be added to Table 4 or 5. It could also be used as part of the uncertainty assessment..*

Correlation Coefficients for each flight have been added in a column under Table 5

*L418f: "Be a bit more specific concerning the near zero differences between model and observation. For the PBL height this statement may be true, however, it is also clear that the model is not able to capture the very sharp increase in potential temperature. I also wonder why the two profiles are shown separately, would be nice to have them on top of each other. Concerning the wind there is a directional shift of about +10 degree in the model. You may call this small, but for the kind of plumes simulated here it might be of importance and should be mentioned."*

I do not discuss the gradient of the potential temperature inversion or the offset in temperature between the model and observations because these factors do not impact the physics of the plume. For now, I would like to keep the two plots separated, as the important thing to emphasize with these plots are the ABL height similarities between the model and the observations which can be seen just fine with the plots as they currently are. Combining the plots would draw the reader's attention away from the ABL height and more towards the 1 K difference in temperature, which is not important to this discussion.

Changed L419-20 to reflect a difference in wind direction of 10 degrees.

*L424ff: Please add a number for the average and maximum enhancement due to upstream natural gas sources at the northern transect to this discussion. From Figure 9 I would think these numbers are around 30 and 80 ppb. It puts the other contributions into perspective.*

Adjusted line in the paper to give a range of the downwind enhancement.

*L473ff: This discussion on the uncertainties due to shifts in the simulated wind direction is very hard to follow. Figure 16 does not speak for itself. You will need to explain in more detail what was done to derive the figure. This discussion should actually given when the cost function is introduced.*

Moved discussion of optimization technique and how it addresses transport uncertainties to methods section. Removed Figure 16 and wrote better explanation as to why sum of squares was not used as optimization technique. Simplified paragraph and associated figure is provided below.

"The decision to use a scalar cost function rather than the sum of squares is to account for possible misalignment between any observed $CH_4$ plume and modelled plumes. There are two potential ways in which misalignment may occur. One possibility is that the modelled wind direction differs from the true wind direction, leading to a plume in the model that is off-centre in relation to the observed plume. The other possibility relates to how the model treats emissions from natural gas as a uniform percent of production. In reality the emissions are more random in nature, and thus the plume may not always develop over the wells with the largest production values. If a cost function is used that minimizes the sum of the squares, any misalignment between the modelled and observed plume will result in the peak of the modelled plume aligning with the height of the tail of the observed plume (Figure 5). Unless the observed plume aligns perfectly with the modelled plume, the optimized emission rate using a sum of squares approach will always bias low. By using a scalar cost function, we solve for an optimized emission rate that results in a plume with the same area under the curve compared to the observed plume (Figure 5). This methodology is not impacted by any misalignment between the modelled vs. observed plumes, preventing the low biases associated with a sum of squares minimization."

[Figure]

Figure 5: (a.) Observed vs model projected CH₄ enhancements during the May 14ᵗʰ, 2015 at 16Z. (b.) Comparison of observed natural gas enhancement to modelled natural gas enhancement along flight path, with upstream emission rate optimized by minimizing the absolute error between the datasets. (c.) Same as previous, but optimized by minimizing the sum of the error between the datasets.

*L486: Should it be wind speed instead of wind direction?*

It should be wind speed. Changed.

L498ff: *"The way the wind speed error is assessed biases could be introduced. The average over all wind observations by the aircraft does not necessarily reflect the average wind speed within the boundary layer, since the sampling is not uniform (especially with height). It seems calculating an average vertical profile of wind speed from all the observations and comparing that to an average model profile generated from the same sampling locations should be more robust. Was this considered and decided that the aircraft sampling was sufficiently uniform?"*

Most of the time the aircraft is flying at a constant altitude. We have about 2-3 vertical profiles (mostly spirals) from the aircraft for most flights, but wind measurements can have large errors while the aircraft rises/falls and turns, both of which occur during a vertical spiral. So we cannot accurately construct a vertical profile of the winds in the ABL using the aircraft data. We could compare the model to surface station measurements in the area, but surface data poorly reflects on model inaccuracies within the boundary layer. Comparing the model directly to the aircraft observations when it is flying steady in the boundary layer is the best method we have of understanding the model's performance throughout the area, and also will capture all of the

potential spatial variability as well. I do acknowledge though that due to a lack of wind measurements with height, if the model has a bias that is not uniform with height this will not be captured in the error analysis.

L506: *"There are more advanced techniques for estimating boundary layer heights from model and observational data than just 'looking' at the potential temperature. For example Bulk Richardson methods are quite useful in situations when potential temperature alone is not providing explicit results. The example shown here is an easy case but I wonder about the other situations mentioned in the text and especially how large the uncertainty of the estimates becomes in such situations."*

Reading the sentence associated with this comment, I may have implied more uncertainty than there actually was. For all 10 flights, the potential temperature inversion could clearly be spotted in the model. This is also true for 9 of the aircraft flights. Only on one day (May 22nd, 2015) did the observational data not clearly show a large potential temperature gradient until ~3000m, with a smaller wiggle in the profile a bit lower. For this one though you could see a quick shift in the all of the trace gases at that wiggle, and so we could conclude that this represented the mixing height. We did not go into a more complicated methodology for deriving the ABL height because we felt it was unnecessary for our cases. Had the potential profiles been messier and vague, we would have then resorted to more complex methods of calculating the ABL height.

I have revised the sentence in the manuscript to make it more clear that there was little ambiguity in finding the ABL height.

L542f: *"How was this done exactly? Did you simply run 3 cases with 1) reference background, 2) background -5 ppb, 3) background +5 ppb? Or did you do a Monte Carlo approach where the +/- 5 ppb could be understood as the standard deviation of a normal distribution? Please clarify and justify the approach also in the light of possible non-linear effects that would not be covered by simply running 3 cases."*

It was done by just running 3 cases to establish the limits because the effects changing the background has on the solution is nearly linear. Unlike using a sum of squares approach in our cost function, the cost function we use tries to minimize the difference between the area under the curve for the observations vs the model. Another way to think of it is that we are minimizing the total enhancement between the sum of the observation-derived enhancements and the sum of the modelled enhancements. By decreasing the background by 1 ppb, we are adding (1 ppb * # of observations) to the total enhancement. By decreasing the background by 2 ppb, we are adding 2 ppb * # of observations) to the total enhancement. The increase (or decrease) is linear, as is the effect on the cost function. The only effect that is non-linear occurs in the process the zeroing of any negative observation-based enhancements (of which you will have more of if you increase the background by 5 ppb), but the impact of this is small on the symmetry of the error bar. So a Monte Carlo approach was not needed due to the simple nature of the errors in this section.

L548f: *It is mentioned that the +/- 5 ppb assumption for the background uncertainty can have different impacts depending on the magnitude of the observed plume and this is related to wind speeds and PBL heights. How can you justify the use of a constant value for the background uncertainty? Shouldn't this also change with meteorological conditions? For example it is mentioned earlier that it will be large when there is more entrainment, which is usually the case while the PBL is still developing, and when plumes from other sources are advected. Wouldn't it make sense to take this variability in the background uncertainty into account?*

In truth, a lot of the changes we see in the background we just don't understand. Below are the methane observations from within the ABL for our flight on May 22nd.

[Figure]

This was a 3 hour flight between 1:30 pm-4:30pm local time. Winds were steady on this day and the boundary layer was deep throughout the duration of the flight. We should expect minimal changes in the background in these conditions. But near the very beginning of the flight (bottom left), values were their largest despite being away from most sources. Meanwhile, values near the end of the flight suddenly drop off by 10 ppb despite having flown over the same area twice just one hour earlier. This is the most extreme example I can present of the randomness of the background variability, but in my experience with aircraft data, there are many instances in which sudden changes in the background can be observed which appear to be unrelated to entrainment or a simple calculation in how much the boundary layer changed. Much of the differences we see in the background values may have to do with heterogeneity of the large $CH_4$ plumes which develop regionally across the U.S.. As a curiosity, we ran WRF-Chem for 1 month at 27 km resolution using the EPA U.S. $CH_4$ emissions inventory (Maakkassers *et al.,* 2016) as input and found that the average difference in enhancement between two locations 71 km away in our study region was 5 ppb. These modelled differences are caused by the spatial

gradients that naturally develop in the methane concentration field due to the transport of methane emissions across the central and eastern U.S.

In short, we picked a background variability of ±5 ppb to represent a conservative estimate of how much variation we could see in a flight. We did not believe that any calculation we could perform with the data we had could sufficiently describe our uncertainty with the background value estimate, so we chose a range that seemed large without being unreasonable.

*L557f: "Again the question: is [the non-natural gas emissions adjustment] done by running by simply running 3 cases or in a Monte Carlo fashion. Did you scale all categories with the same factor at the same time or test combinations as well? How do you obtain symmetric uncertainties in the final emission rate when you use asymmetric uncertainties in the non-upstream natural gas emissions?"*

The 50-200% range was a typo and should have read 0-200%. However, we agree that the initial error analysis from the inventory could be improved upon. To improve this section, we followed the advice provided and ran a Monte Carlo simulation for 10,000 iterations, selecting a multiplier from a uniform distribution ranging from 0 to 2 for each of the 8 non-upstream gas sources. This allowed for a much more realistic understanding of the errors compared to assuming all sources were incorrect by the same multiplier. The resulting patterns for each day were close to Gaussian. For each day, we fit a Gaussian and took the 2 sigma range as the uncertainty. Most days did not see much of a change from the previous uncertainty assessment, but days where the coal plume interacted with the observations (May 24th May 25th) saw an increase in the error. The methodology for this has been updated in the paper.

*L505, L562: "Table 5 should already be referenced here and not only at the very end of the section."*

Added for L562. I didn't add a reference at L505. That's describing a different methodology related to Table 4. I did add another reference to both Table 4,5 when the error analysis is first introduced.

*L571: "The estimator for the model performance uncertainty is a normalised root mean square error (NRMS) and it should be called like this to reduce the confusion. However, I am not convinced that this is the best estimator for what you want to achieve. One problem is the use of the optimised simulation, which means you have already used the information contained in the observations and thus the uncertainty estimator is not*

*independent anymore. Furthermore, NRMS is not a good measure of what you call "similarity of pattern". The Pearson correlation coefficient should be better suited for this purpose. Please justify your choice. If you stay with a RMS estimation it should be applied to the prior simulations and the bias should be removed before the RMS is estimated."*

We had originally looked at the correlation as a way to quantify uncertainty, but we found there were some interesting cases where the correlation did not accurately describe problems with the model vs. obs comparison. For a great example, look at Figure 17 (May 22nd Loop1 vs May 28 Loop2).

[Figure]

Figure 17: Comparison of observed natural gas enhancement to modelled natural gas enhancement for segments along the (left) May 22nd flight and (right) May 28th flight. A distinct lack of representativeness of the observations in the modelled enhancement can be seen in the May 22nd flight compared to the May 28th flight.

Anyone looking at the figures would immediately conclude that the observations on May 28th were modelled better than May 22nd. However, the May 22nd Loop1 flight actually has a correlation coefficient of 0.58 while the May 28th Loop2 flight has a correlation coefficient of 0.57. Of all the methods we tried, we found the NRMS method we use resulted in the error estimates that matched with a common-sense approach.

 I do not fully understand the request to apply our methodology to the prior. Unlike an atmospheric inversion, our forward-based modeling experiment does not have a true prior. The number we selected for our first-guess emission rate (0.13% of production) has 0 bearing on the final results. It only provides the model with the spatial pattern the plume will take, and is scaled entirely by the observations. Had we started with a

first-guess emission rate of 82,350% of production, the final result and uncertainties would not change.

Added a line addressing that the formula used is a modified NRMS to avoid confusion.

*"L579f: You also assume Gaussian uncertainty distributions, which seems oversimplified."*

As addressed above, the certainty of our background uncertainty parameter is very uncertain. The Monte Carlo for the non-natural gas emissions uncertainty produced mostly Gaussian-shaped distributions, but this was assuming an arbitrary range of variations in the emissions of each source which may also have spatial variabilities that we don't understand, and the model performance error is just a method we picked to try and best match our own understanding of what the errors associated with mismatch should look like. Assuming Gaussian uncertainty distributions is probably a stretch, but there was no methodology we could come up with where we would feel any more confident in our final error bounds.

To counteract the uncertainty in our uncertainty, it should be noted that we tried to be very conservative with our uncertainty parameters. A ±5 ppb variation with the background is large. A 0-200% range in the emissions from non-natural gas sources is large. We did not want to underestimate the uncertainty, as is often done, so we tried to be as fair with our analysis as possible to produce a 2σ range which would contain the true emission rate 95 times out of 100.

*L645:" Figure 18 can go into a supplement without any loss of information. Instead give the rate of change of the background with time as a number in the text."*

Added the rate of change in the text, and Figure 18 is in consideration to be moved to supplemental (see major comments below)

*L652: "Does this mean the final uncertainty estimate is the mean over the uncertainties for each flight?"*

From that sentence, we use the standard error from the four flights to come up with the uncertainty range.

*L663ff and Fig. 19: "The variability in emission rate estimates for individual days as shown in the figure are not discussed in any detail.*

This is a good point. Added a segment in the early part of the discussion section to discuss reasons for variability in the emission rate.

*Of special interest is the question of agreement between the two methods as it could lend some additional trust in the applied methods. However, the figure seems to raise more questions in the reliability. For example if 29 May is the golden day (as described earlier in the text), why are estimates from both methods so different?*

[Figure]

**Figure 19: Calculated upstream natural gas emission rates using (black) model optimization technique and (red) aircraft mass balance technique.**

The values on May 29th from the aircraft mass balance and model optimization are 0.57% and 0.35%, differing by 0.22% of production. They even fall within each other's error bars. Claiming the two values are "so different", is odd, and should not be used to question the reliability of the model optimization method. Furthermore, the aircraft mass balance method has sources of error which differ from errors associated with the model optimization technique. A perfect match between two methods each with their own unique sources of error is unlikely.

I do discuss briefly in the paper that the relative closeness of the results in these two methods gives credence to the model optimization technique. But that is not to say the aircraft mass balance method is the golden standard in the field. If the mass balance

approach were flawless, bottom-up and top-down emission estimates would be in agreement at this point and there would be little need for this paper.

*Also there seems to be no agreement in the variability for the 4 common days. How could you explain the large variability from the point of the emission processes? Is there any dependency on the total production on these days (following the line of thought used later for the discussion of different basins)? Please speculate on this otherwise the reader is left with the impression that your given uncertainty estimate is rather optimistic."*

The topic of emissions variability is briefly touched upon in the introduction, but I've added a segment in the discussions to re-address some of the possible reasons for the variability in the totals. These reasons include the nature of natural gas releases from stochastic events such venting, flaring, liquids unloading, ect., as well as the possibility that the days with large emission rates also have large uncertainties due to the complex scenarios from those days.

*L666f: Repeat the range of emission rates from other studies here.*

Added a line addressing this rate compared to other regions.

*L696ff: The flight described in this section and shown in Fig. 21 is not part of the main analysis of the paper and should be omitted as it does not provide any additional insights and aspects that have not or could not be shown on the other flight examples.*

On the contrary, this flight is the perfect example to demonstrate why it is so advantageous to model the plumes with a direct model instead of the backward adjoint model. Until this study, a mass balance from a single flight was the only top-down study performed in the northeastern Marcellus. There was little knowledge of how the coal plumes 500 km away would interact with the flight in this region. By running a transport model, we're able to see that the background of this flight is compromised by the heterogeneous nature of the coal plume, thus explaining the complex signals observed by the aircraft (it should be noted that the highest $CH_4$ values observed in *Peischl et. al.* 2015 were **upwind of the wells)**. Even if someone does not plan to use the transport model to solve for the emissions in a region, it can still be useful to better understand the complexity of background plumes on a given day and decide whether a mass balance would be appropriate to use. By using the flight from *Peischl et. al.* (2015), we not only show the usefulness of using a transport model for future studies, but also emphasize the potential need to relook at flights from earlier studies and check for

whether complex plume structures may be adding additional uncertainty to the regional emissions estimate.

**Addressing the major concerns directly**

*Model optimisation technique (inverse modelling): The applied optimisation technique uses a very simple cost function to find an optimal factor between 'prior' and 'posterior' emission rates. Why did you not apply a Bayesian approach that could take uncertainties in the prior values and observations/simulations into account? Much of the following uncertainty assessment could then be used in the optimisation step and also more reliably give posterior uncertainties. Also, why do you not use sum of squares in the cost function as is commonly done? Some explanation is given later in the text, but without further discussion of Figure 16 this remains useless. Please discuss why your method should provide more reliable results as other similar studies that have used a Bayesian framework.*

As far as I'm aware, there are not any similar regional methane emission studies that have combined a Bayesian framework with an aircraft campaign. We know of few ongoing studies but to date, none of them have been published. Few top down studies used gas-to-gas ratios and backward footprints to estimate point source $CO_2$ emissions (e.g. Brioude et al., 2012) but without any prior emissions. One of the advantages of a Bayesian methodology is that it allows one to use and propagate their errors from a prior guess to the posterior estimate. However, it is for that same reason that a Bayesian approach would be inappropriate here. We have little information regarding the errors associated with this study. The background $CH_4$ values, the emissions inventory, and the model transport error are all poorly understood. This includes the 'prior' emissions estimate of natural gas as well, which is just the median emission rate from 17 UNG wells in southwestern PA. Any Bayesian approach without reliable uncertainty estimates will not produce reliable results (Chevallier et al., 2006).

One of the greatest advantage to our forward-based modelling approach is that you are able to visualize plumes and partition the problem into components, directly observing the uncertainties associated with the optimization process. The forward modelling approach lets you directly see the structures of the different plumes on a given day and how they compare to observations. This makes it simple to know your problems for a specific day. If there are far-reaching plumes influencing the observations, you know to be cautious with the emissions estimate. If plumes appear to consistently misalign slightly between the observations and the model, you know there's likely a transport issue. If observations shift in regions where no plumes are modelled, you know you are

dealing with either a missing source or background variability. And because emissions scale linearly with the resulting plumes, once the model is run it becomes easy to make adjustments to see how much impact changes can have on your results. These different aspects are purely and simply invisible in a backward mode, for which the adjoint only points to the simulated area of influence. The difficulty with the direct approach is to rigorously quantify the final uncertainties, but uncertainties obtained from a Bayesian will be no more reliable if the prior uncertainties are not known.

So while the approach we use in this study may be simplistic, we do not consider that to be a weakness. Rather, the ease-of-use associated with this method allows it to be readily applied to any other study, past, present, or future, so long as an emissions inventory can be compiled for the region.

We have responded to the "sum of squares" issue under minor comments. We have considered methods such as Nearest Neighbor Search (NNS) but found it tedious and unsatisfactory as it would look for surrounding pixels and would therefore require an adjoint model.

*Uncertainty assessment of obtained emission rates: The uncertainty assessment of the obtained emission rates for both the 'model optimisation' as well as the mass balance method contains a lot of arbitrary assumptions and does not seem to be statistically sound. Their seems to be an overall assumption of Gausssian uncertainties (although not explicitly mentioned), which does not seem to be justified for several of the sources of uncertainty. Also the way the uncertainties are propagated through the individual methods remains very vague and could be improved by using a Monte Carlo type uncertainty assessment. These concerns are repeated in detail below.*

We addressed this concern at multiple points under minor comments section. To summarize here, we have propagated errors when possible in the revised manuscript (non-Natural Gas sources), and found that our Gaussian assumption matches closely the exact error distribution in this case. For the other error terms, we have statistical information on transport errors (from other studies at similar scales, e.g. Deng et al., 2017) to confirm it, or reasons to use a Gaussian propagation of the standard deviations when the distribution remains unknown (e.g. prior errors on emissions). We also want to emphasize that the absence of known prior uncertainties for emissions will significantly impair our ability to diagnose the full error distributions with a Monte Carlo approach, Therefore, random sampling such as Monte Carlo would not solve the problem of unknown error distributions. In general, we have been very conservative in our estimates, using the high end of the error values. Unless more information on error

sources becomes available, we think that propagating errors under Gaussian assumptions remains valid and perfectly justified.

*Length: With 21 figures, 8 tables and a total of 50 draft pages the manuscript is quite lengthy and it would profit from shortening and restructuring. Below I suggest a number of figures that could easily be omitted or moved into a supplement without loss of information. Furthermore, I strongly encourage to incorporate the description of the uncertainty assessment method, which is now given within the results section, into the methods section along with or following the description of emission rate estimation itself. In this was many repetitions could be avoided and the paper could really focus onto the results in section 3.*

A major restructure was done to move uncertainty assessment into methods section and eliminate duplications associated with having those split previously. Results are now concise and focused only on results, rather than methods. The number of figures has been reduced by combining Figures 1 and 20, and the removal of Figure 16. If the editor feels that the length of the paper is an issue, further changes can be made, such as moving Figure 4 and 18 to supplemental as well as the entire description of the mass balance method used in this paper. However, if the editor feels that length is not an issue, we would prefer to keep these sections in the paper to retain its flow.

**Technical Fixes**

*L26: Abbreviation (WRF-Chem) not introduced before. Do so on line 25.*
Added abbreviation

*L48: "transforms" instead of "transform"*
Due to interactions… which transform. Grammar is correct

*L59: Add "United States" in front of "Environmental Protection Agency".*
Added

*L348: Why are the terms U, D, and L given in braces? Not necessary.*
Braces removed.

*L496: "as long as" instead of "so long as".*
Changed

*Fig. 7: Do not show wind arrows in center panel. Not visible anyway and one gets the general flow direction from the shape of the plumes. Show center panel with same size as the individual tracers. Also use exactly the same color scale for all sub-panels. It looks like the center plot does have fewer color levels than the others. Each panel (also the center panel) requires a label.*

We respectfully disagree with this comment and prefer to keep the wind arrows to illustrate the direction of the flow for this very specific case. The coal plume coming from far away is explained by the regional circulation. The arrows remain small and should not decrease the readability of the figure. We modified the colorscale to match better with individual tracer figures. Did not resize middle figure to match other figures. Middle figure is most important and thus should be drawn as such. I think the label being referred to each panel needing was lat/lon. Added to the main figure. Not added to smaller figures as it makes things messy and unreadable. I would think the reader would have a good understanding of where they are in space using middle figure as reference.

[Figure]

*Fig. 11: Show observed and simulated profile on top of each other (see also comment above).*

The purpose of the figure is to show an example of a theta profile for both the observations and the model and how the abl was found for each. The current figure accomplishes this fine.

[Figure]

*Fig. 12 and others: There is no mentioning of the wind arrows in the figure legend. Are these near surface or mean PBL winds? There is also no reference vector that would allow inferring the wind speeds from the arrow length. Choose a different color for the wind vectors. They are not well visible in the righthand figure.*

Added description of height level of model used in figure. Wind speed by arrow length changes in a complex manner from figure to figure with no easy way to retrieve its scale. Given the context of the figure, the reader will likely not be interested in small changes in the wind speed but rather the wind direction. If a reader wants more information on the wind speed, he can reference Table 3 which contains information on the mean wind speed for each flight. If the editor feels that a wind speed legend is necessary for these figures, then they will be added in. In terms of wind arrow color, it is difficult to pick a color that works everywhere due to the various colors on the plots associated with the plumes. For figure 12 this should not be an issue as it is a two panel figure where wind directions are clearly visible on the left panel. In figure 14 the wind direction is consistent throughout the domain and should not be too difficult for the reader to extrapolate the information.

*Fig. 19: Mention in the caption what error bars represent.*

Added description.

*Fig. 20: This information could also be merged into Fig. 1 and save the additional figure here.*

Merged well production information from Figure 20 into Figure 1. Removed Figure 20 from the paper.

[Figure]

**Other Changes**

Fixed error in Corrected NG Emission Rate (Table 4: WS/ABL bias table), where errors were miscalculated.

Fixed error in calculation of mean and uncertainty regarding aircraft mass balance emission rate.

Minor text fixes.

---

## Author Response (AR2)

**Response to Reviewer/Editor**

-The author agrees with the reviewer that Figure 4 is redundant given existence of other figures which show study region, and was not worthy of a supplemental section. It has been removed from the paper.

-The reviewer correctly spotted a mistake in the written emission totals in the first sentence of the discussion section. This has been fixed to match text elsewhere.

-An adjustment was made to the department of one of the co-authors.

-A sentence was added in the acknowledgement section to acknowledge the gratefulness of the author for their hard work catching the many stupid and smart mistakes scattered throughout the paper.

[revised manuscript text omitted]